# Microglia-neuron interaction at nodes of Ranvier depends on neuronal activity through potassium release and contributes to remyelination

R. Ronzano [1,5], T. Roux[1,2,5], M. Thetiot[1], M. S. Aigrot[1], L. Richard[3], F. X. Lejeune[1,4], E. Mazuir [1], J. M. Vallat[3], C. Lubetzki[1,2,5] & A. Desmazières [1,5 ✉]

Microglia, the resident immune cells of the central nervous system, are key players in healthy brain homeostasis and plasticity. In neurological diseases, such as Multiple Sclerosis, activated microglia either promote tissue damage or favor neuroprotection and myelin regeneration. The mechanisms for microglia-neuron communication remain largely unkown. Here, we identify nodes of Ranvier as a direct site of interaction between microglia and axons, in both mouse and human tissues. Using dynamic imaging, we highlight the preferential interaction of microglial processes with nodes of Ranvier along myelinated fibers. We show that microglia-node interaction is modulated by neuronal activity and associated potassium release, with THIK-1 ensuring their microglial read-out. Altered axonal $K^+$ flux following demyelination impairs the switch towards a pro-regenerative microglia phenotype and decreases remyelination rate. Taken together, these findings identify the node of Ranvier as a major site for microglia-neuron interaction, that may participate in microglia-neuron communication mediating pro-remyelinating effect of microglia after myelin injury.

[1] Sorbonne Université, Paris Brain Institute (ICM), INSERM U1127, CNRS UMR 7225, Hopital Pitié-Salpétrière, Paris, France. [2] Assistance Publique des Hôpitaux de Paris (APHP), Hopital Pitié-Salpêtrière, Département de Neurologie, Paris, France. [3] Centre de Référence National des Neuropathies Périphériques Rares et Département de Neurologie, Hopital Universitaire, Limoges, France. [4] Paris Brain Institute's Data and Analysis Core, University Hospital Pitié-Salpêtrière, Paris, France. [5] These authors contributed equally: R. Ronzano, T. Roux, C. Lubetzki, A. Desmazières.. ✉email: anne.desmazieres@icm-institute.org

Microglial cells are the resident immune cells of the central nervous system (CNS), where they represent 5–10% of the cells[1]. This is an heterogeneous population, which participates in normal brain development, homeostasis, and maintenance of neuronal function, as well as learning and memory, by modulating neurogenesis, neuronal survival, wiring, and synaptic plasticity[2–4].

In the healthy brain, microglia dynamically survey their environment with their motile processes, as well as with nanoscale sensing filopodia[5–7]. Process motility is modulated by ATP, chemokines, neurotransmitters, extracellular potassium concentration[5,8–10], among other cues, and integrated through microglia receptors, in particular the purinergic P2Y12 receptor and the fractalkine receptor CX3CR1, and the recently identified two-pore domain potassium channel THIK-1[10–12].

Microglia are activated in most neurological pathologies, including neurodegenerative diseases, epilepsy, autism, psychiatric disorders, and stroke[2,4]. In multiple sclerosis (MS), a CNS inflammatory, demyelinating and neurodegenerative disease, activated microglia can contribute to neuronal loss, but they are also important to favor myelin regeneration, in particular through removal of myelin and neuron debris[13–16]. Remyelination depends on the phenotype of activated microglial cells, with a polarization from M1 (pro-inflammatory) to alternatively activated M2 (pro-regenerative) microglia, though this vision of a M1/M2 polarization is clearly oversimplified[17–20]. The pro-inflammatory signature is observed early following injury and is associated with deleterious microglial activity if maintained inappropriately[14,21,22]. However, cells with a pro-inflammatory signature also stimulate proliferation and recruitment of oligodendrocyte precursors cells (OPCs) towards the lesion, whereas pro-regenerative microglial cells promote OPCs differentiation into myelinating oligodendrocytes[14–16,22]. These experimental data are consistent with results showing an enrichment in pro-regenerative microglia in remyelinating lesions[14–16].

It is established that microglia sense neuron activity and can modulate neuron function or detect early neuronal damage[9,11,23–29]. Aside from neuronal synapses, neuronal soma has recently been identified as a site of microglial interaction, relying on purinergic signaling and possibly involved in microglia-induced neuroprotection[11,28,29]. The axon initial segment (AIS) has also been identified as a site of microglia-neuron contact, with description of microglial processes overlapping AIS in healthy brain, which vary following brain injury and inflammation[30,31].

Along myelinated fibers, the nodes of Ranvier are short unmyelinated domains allowing action potential regeneration and propagation. Astrocytic processes have been described to contact nodes of Ranvier, with a role in nodal length modulation and in ionic buffering[32]. OPCs processes also contact nodes, although the prevalence of these contacts as well as their physiological role remain elusive[32]. Direct contacts of microglia on myelin sheaths and nodes of Ranvier have recently been observed in rat corpus callosum[33]. We therefore tested, in control and demyelinated CNS, the hypothesis that nodes of Ranvier might be a preferential site for axon-microglia communication.

Here we identify nodes of Ranvier as a site for microglia-neuron communication, in mouse and human CNS, suggesting they could play the role of a "neuro-glial communication hub". We show that microglia-node interactions are modulated by neuronal activity and potassium ion release, and provide evidence of their influence in microglia-dependent remyelination capacity after experimental demyelination.

## Results

**Microglia contact nodes of Ranvier in vivo in mouse and human tissues.** The nodes of Ranvier, short unmyelinated axonal domains, allow direct access to the axonal surface of myelinated axons. We thus addressed whether microglia can contact axons at the nodes of Ranvier.

By performing immunostainings and 3D reconstruction on adult mouse fixed tissue, we first observed that microglia (Iba1+ cells) contact nodes of Ranvier (AnkyrinG+ structures surrounded by paranodal Caspr+ staining) in CNS gray (cortex and cerebellum, Fig. 1Ai–ii) as well as white matter (corpus callosum and cortico-spinal tract, Fig. 1Aiii–iv), thus confirming and extending a previous observation in rat corpus callosum[33]. The vast majority of microglial cells contact multiple nodes, through cell soma or processes (at the tip or "en-passant"), with the preferential area of contact being along the node and extending to the junction of the node and the paranode (Fig. 1A). This was confirmed by an electron microscopy study following immunostaining with Iba1 antibody, to visualize microglial processes. Longitudinal views of nodes delineated by the paranodal loops on both sides and contacted by microglial processes show that these processes directly contact the nodal axolemma (Fig. 1B) and that the microglial process sometimes extends towards the first paranodal loops (Fig. 1Bii–ii').

Knowing that nodes can also be contacted by astrocytes and OPCs, we next addressed whether a single node could be contacted by multiple glial cell types. We performed co-stainings in adult mouse spinal cord tissue for microglia (Iba1) and astrocytes (GFAP, Fig. 1C) or OPCs (PDGFRα, Fig. 1D). Our results, showing that a single node can be contacted simultaneously by multiple glial processes, suggest that the node of Ranvier could be a neuro-glial "hub".

Microglia-node contacts were further detected in human post-mortem hemispheric white matter of healthy donors (Fig. 2), as shown using a specific marker for resident microglia (TMEM119, Fig. 2A, B) and a marker of homeostatic microglia (P2Y12R, Fig. 2C), with 20% of the nodes contacted by microglia (Fig. S1).

**Microglia-node of Ranvier contacts are durable and their frequency increases during remyelination.** To assess the extent of microglia-neuron contacts at nodes, we first quantified the percentage of nodal structures contacted by microglial cells in adult mouse dorsal spinal cord. As shown in Fig. 3B, E, 26.3 ± 3.7% of nodes are contacted by microglia. To assess whether these contacts correspond to microglial cell random scanning of the local environment or to a directed and/or controled interaction, we then quantified the stability of the contacts, using in vivo live-imaging of mouse dorsal spinal cord[34] from CX3CR1GFP/+/Thy1-Nfasc186mCherry double-transgenic mice allowing to detect both microglia (GFP) and nodes of Ranvier (Nfasc186mCherry; Fig. 3H, I). We confirmed that microglia are dynamic cells, with motile processes and that their whole morphology can vary over long periods of time (Movie 1). However, when selecting nodes initially contacted by microglia (Figs. 3J, S2A–C, and Movies 1–4), we observed that the microglia-node contacts were maintained along the vast majority of 1-h movies, as well as 3-h movies (Figs. 3K, L and S2D–E respectively). This suggests that microglia-node contact is a specific interaction rather than a random scanning.

We then used the same methods to assess microglia-node interaction following a demyelinating insult (focal demyelination induced by lysophosphatidylcholine, LPC, Fig. 3A). As nodes are deeply disrupted by demyelination[35], we excluded demyelinated areas and instead analyzed the perilesional area (i.e., the periphery of a demyelinated lesion, where nodal structures are preserved despite paranodal abnormalities; 7 DPI). Remyelinating lesions (11 DPI) and corresponding Shams, injected with LPC carrier (NaCl 9‰), were also analyzed (Figs. 3C–G and S2).

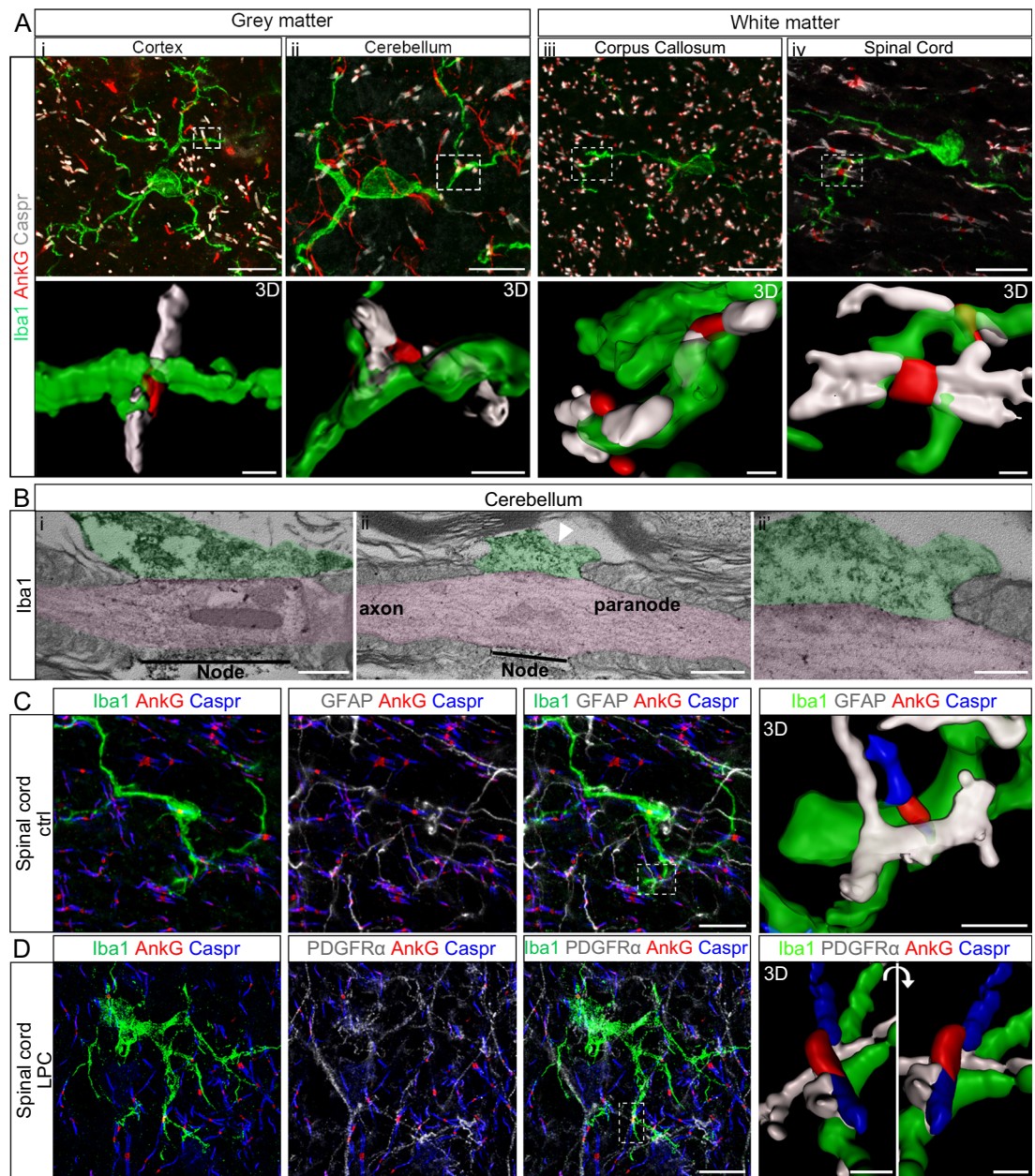

**Fig. 1 Microglial cells contact nodes of Ranvier in mouse central nervous system. A** In adult mouse nervous tissue, microglial cells (Iba1, green) contact nodes of Ranvier (AnkyrinG, indicated as AnkG in the figures, red) in grey (i, cortex and ii, cerebellum) and white matter (iii, corpus callosum and iv, dorsal spinal cord, cortico-spinal tractus). **A**i–iv 3D reconstruction of boxed area in i to iv respectively. **B**i–ii' Transmission electron micrographs showing microglial processes (Iba1+, green) contacting directly the nodal axolemma (pink) in adult mouse cerebellum. ii' Higher magnification of the micrograph (ii), showing the interaction between the microglial process, the axolemma and first paranodal loop. **C, D** Immunofluorescent stainings of adult mouse dorsal spinal cord showing nodes of Ranvier (AnkyrinG, red) contacted by both a microglial cell (Iba1, green) and an astrocyte (GFAP, white) in control condition (ctrl) (**C**) and a microglial cell (Iba1, green) and an oligodendrocyte progenitor cell (PDGFRα, white) in remyelinating condition (LPC) (**D**). 3D reconstructions correspond to the boxed area. Scale bars: **A, C, D** 2D: 10 μm; 3D: **A** 1 μm, **C, D** 2 μm; **B**i–ii 500 nm, **B**iii 200 nm. **A, C, D**: $n = 4$ animals, **B**: $n = 3$ animals.

We found that the frequency of interaction (i.e. the percentage of nodes contacted by microglia) was unchanged in the perilesional area (perilesional: 37.6 ± 2.7% vs sham: 31.8 ± 3.5%; Fig. 3C–E). In contrast, the stability of the interaction was significantly decreased in perilesional tissue at the peak of demyelination (1-h movies, 88.4 ± 3.1% of timepoints with contact) compared to control and 7 DPI Sham (98.9 ± 1.1% and 100 ± 0.0% respectively, Fig. 3K). Similar results were obtained with the longest sequence of consecutive timepoints with contact, with a reduction of 25% of contact stability (Fig. 3L). In contrast, the

frequency of interaction was significantly increased during remyelination (60.9 ± 1.9% of nodes of Ranvier contacted; Fig. 3E, G) compared to control and 11 DPI Sham (18.2 ± 1.6%; Fig. 3E, F), which was not due to changes in microgial cell or node numbers (see statistical analysis table). Furthermore, there was a restoration of long-lasting interactions during remyelination, similar to controls (1-hour movies, Figs. 3K, L and S2 Movies 2–4). The 3-h acquisition study showed similar tendencies (Fig. S2D–E).

The increased frequency of interaction upon remyelination suggests that microglia-node communication might play a role in

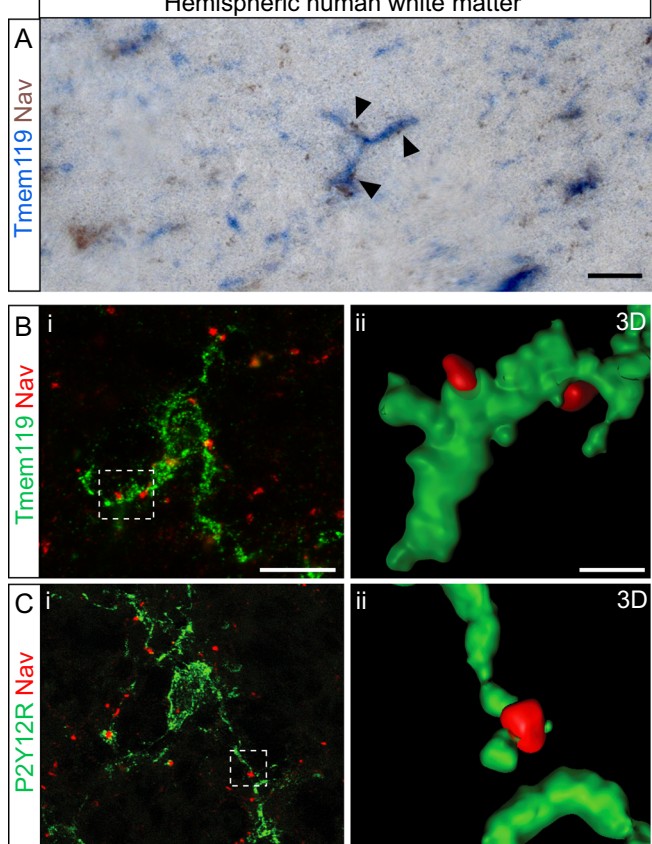

**Fig. 2 Microglial cells contact nodes of Ranvier in human central nervous system. A** Immunohistostainings of post-mortem human hemispheric white matter tissue (healthy donor) showing microglial cells (blue, resident microglia, TMEM119) contacting nodes of Ranvier ($Na_V$, brown). **B**, **C** Immunofluorescent stainings of post-mortem human hemispheric white matter tissue (healthy donor) showing microglial cells (green; **B**, resident microglia, TMEM119, and **C**, homeostatic microglia, P2Y12R) contacting nodes of Ranvier ($Na_V$, red). **B**ii, **C**ii 3D reconstructions correspond to the boxed area in **B**i and **C**i respectively. Scale bars: (2D) **A**, **B**, **C** 10 μm; (3D) **B**, **C** 2 μm. **A**: $n = 6$ samples, **B**, **C**: $n = 5$ samples.

the repair process, whereas the decreased stability in the perilesional area (where demyelination is not yet detected) might be an early event preceding demyelination.

**Microglial processes interact preferentially with the nodes of Ranvier compared to the internodes.** We next studied microglia-node interaction over shorter periods of time to determine whether the contacts were stable along time or could correspond to back and forth microglial contacts. Microglial cell bodies and thick processes contacting nodes were mostly non moving during short periods of time and we thus focused on contacts between nodes and the thin microglial process tips, which are very motile. We in particular wanted to address whether the contact at nodes by these processes corresponded to a random screening of the environment or whether they would preferentially interact with the nodal area. To obtain the increased temporal and spatial resolution needed to visualize microglial thin process tips and to explore this question in myelinated as well as (re)myelinating contexts, we used organotypic cultures of mouse cerebellar slices, a model which presents a preserved cytoarchitecure and is highly accessible for spinning-disk confocal live-imaging[36,37]. It is

further adapted for electrophysiology studies and pharmacological treatments.

In fully myelinated $CX3CR1^{GFP/+}$ slices (Fig. S3F), the percentage of nodes of Ranvier contacted by microglia is comparable to in vivo (23.9 ± 4.0%, Fig. S3H).

Microglia also contact nodes of Ranvier during developmental myelination, both in vivo and ex vivo (Fig. S3B, C, respectively). In myelinating tissue ex vivo, the percentage of nodal structures contacted is similar to adult myelinated tissue (21.8 ± 3.4%, Fig. S3D), with no significant bias towards mature nodes of Ranvier or immature nodal structures (node-like clusters and heminodes, Fig. S3A, E). Following LPC treatment, and in agreement with our in vivo observations, the percentage of nodal structures contacted in remyelinating slices (Fig. S3G) is increased compared to control condition (46.0 ± 3.2%, Fig. S3H), without any bias towards mature or immature structures (Fig. S3I).

Having established the similarity of ex vivo and in vivo models for the analysis of microglia-node interaction, we then used $CX3CR1^{GFP/+}$ cerebellar slices transduced to visualize nodal structures ($\beta1Na_V$-mCherry neuronal expression, Fig. S4A, B) along Purkinje cell axons (10-min movies; Fig. S4C, D and Movies 5–7), to gain deeper insight into the dynamics of microglia processes at nodes. We first compared microglial process tip dynamics when contacting an internodal or a nodal area along the axon in myelinated slices (Fig. 4A, Control, Movies 5 and 6, respectively). The rate of microglial process contact is significantly increased at nodes compared to internodes (81 ± 8% of timeframes per movie with contact at nodes, vs 52 ± 5% at internodes, Fig. 4B) as well as the maximum number of consecutive timeframes with contact (14.0 ± 2.2 frames at nodes vs 8.2 ± 1.1 frames at internodes, Fig. 4C). Similarly, the number of consecutive timeframes without contact is increased at internodes (8.5 ± 1.2 frames vs 3.9 ± 1.6 frames at nodes, Fig. S4E). Regarding microglial contacts at nodes, these parameters are similar in remyelinating slices (rem) compared to control slices (Fig. 4A, F, G, Movie 7 and Fig. S4G).

**Microglial process behavior is modified at the nodes of Ranvier.** To further assess how microglial process motility and behavior could be modified at the vicinity of a node, we further analyzed the trajectory of process tips with nodal contact or without contact (wo contact) at imaging onset (Fig. 4D, E). The total length of the trajectory was unchanged along 10 min (wo contact 28.1 ± 4.0 μm vs nodal contact 26.1 ± 4.4 μm, Fig. 4D), as well as the mean instantaneous velocity (wo contact: 2.6 ± 0.4 μm/min vs nodal contact: 2.2 ± 0.4 μm/min, Fig. 4E). However, the microglial process tip remained in the direct vicinity of the node contacted compared to non-contacting tips, which frequently move away from their initial position (mean distance from origin, wo contact: 3.06 ± 0.4 μm vs nodal contact: 1.77 ± 0.4 μm, Figs. 4D and S4F).

In remyelinating slices, the mean distance from origin was also reduced for processes contacting nodes (wo contact: 2.88 ± 0.7 μm and with nodal contact: 1.20 ± 0.5 μm, Figs. 4H and S4H). Interestingly, the mean velocity was in that case significantly decreased by a half for process tips contacting a node compared to non-contacting ones (wo contact: 2.1 ± 0.3 μm/min vs node: 1.2 ± 0.3 μm/min, Fig. 4I). Hence, the total length of the trajectory covered in 10 min was decreased when contacting a node (wo contact: 24.9 ± 3.7 μm vs node 15.8 ± 4.2 μm).

Taken together, these data suggest the existence of (an) underlying mechanism(s) attracting and/or stabilizing microglia at nodes of Ranvier, including the very motile microglial processes. Microglial interaction at the node is further

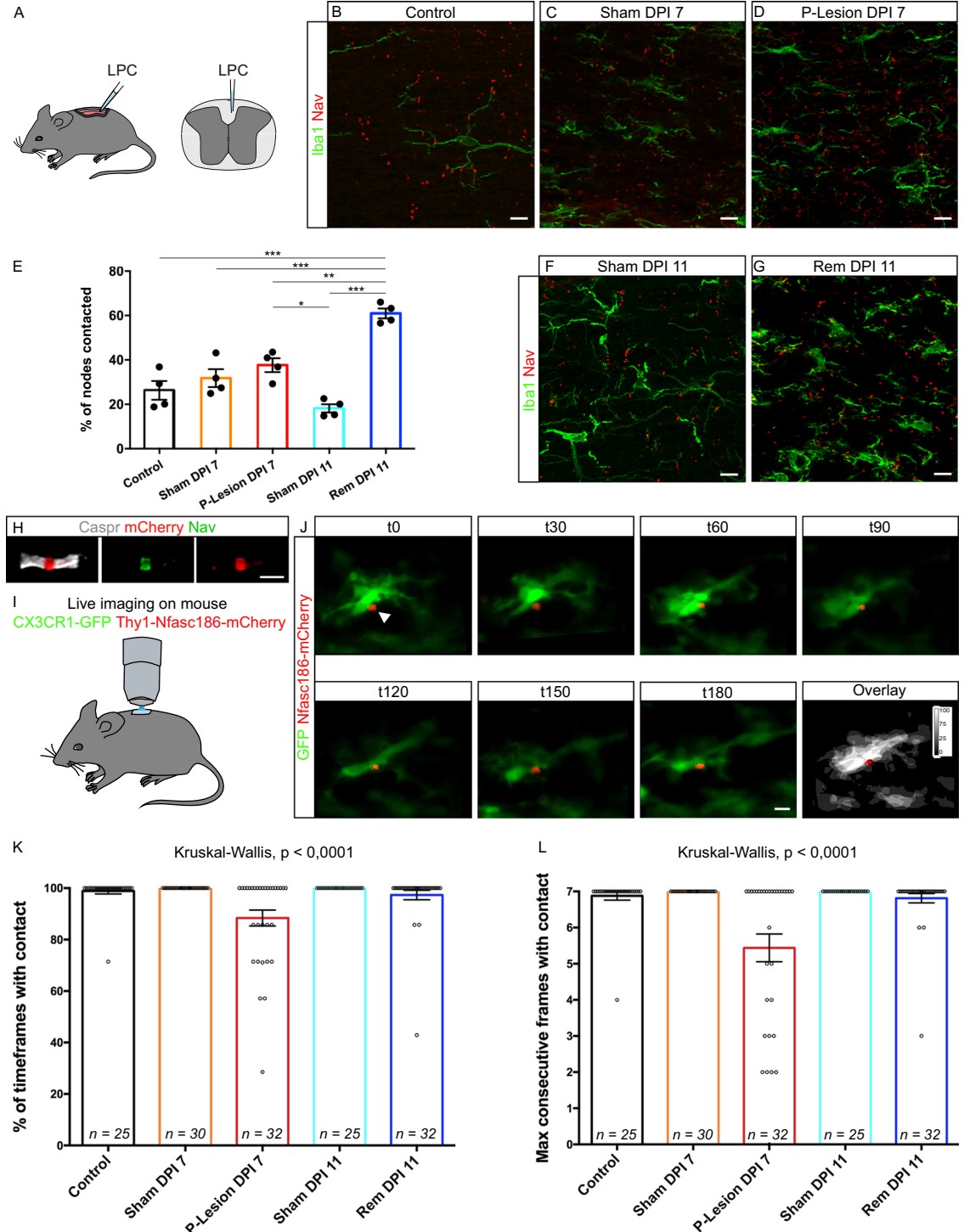

**Fig. 3 Microglial cell contacts at nodes of Ranvier are stable in vivo and increase in remyelination. A** LPC injection in mouse spinal cord (cortico-spinal tractus). **B–G** Contacts are observed between nodes of Ranvier (Na$_V$, red) and microglia (Iba1, green) in control (without injection) (**B**), Sham 7 and 11 days post-NaCl injection (Sham DPI 7 and 11) (**C**, **F**, respectively), in perilesional tissue 7 days post LPC injection (P-Lesion DPI 7) (**D**) and in remyelinating tissue 11 days post-LPC injection (rem DPI 11) (**G**). **E** Corresponding quantifications of the percentage of nodes contacted. **B–G**: $n = 4$ animals per condition; 4–6 areas per animal, 42 nodes minimum per area). **H** Nfasc186mCherry colocalizes with Nav at the node of Ranvier in Thy1-Nfasc186mCherry mouse dorsal spinal cord ($n = 3$ animals). **I** Glass-window system above dorsal spinal cord for 2-Photon live-imaging. **J** Images from a 3-h movie (Movie 1) from a CX3CR1-GFP/Thy1-Nfasc186mCherry mouse shows a stable interaction between a microglial cell (green) and a node of Ranvier (red) (arrow head). scale bar: 10 μm. **K** Percentage of time in contact in 1-h movies, with one acquisition every 10 min. **L** Longest sequence of consecutive timepoints with contact in 1-h movies. Each dot is a microglia-node pair. The number of node-microglia pairs imaged is indicated on each bar. **J–L** $n = 6$–11 animals per condition. Scale bars: (**B–D**, **F**, **G**, **J**) 10 μm. **E** ANOVA with post hoc Tukey test; **K**, **L** Two-sided Kruskal–Wallis test. $*P < 0.05$, $**P < 0.01$, $***P < 0.001$, $****P < 0.0001$, ns not significant. Bars and error bars represent the mean ± s.e.m. For detailed statistics, see Supplementary Table.

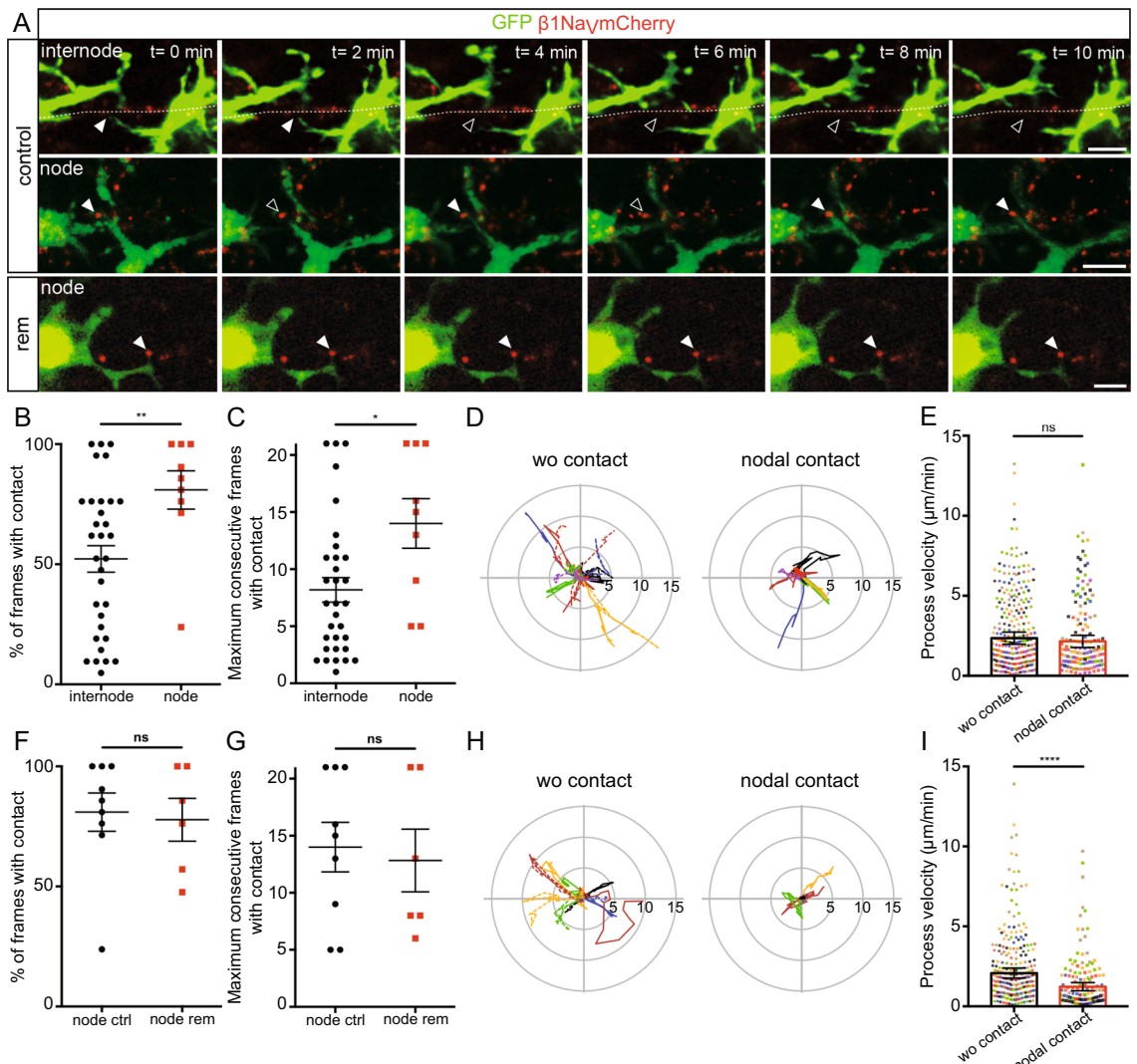

**Fig. 4 Microglia process dynamics are modified by nodal structure vicinity in myelinated and remyelinating cultured slices. A** Microglial cells (green) initially contacting an internode or a node (red) in myelinated slices (control), or a node in a remyelinating slice (rem). Arrowheads show the initial contact position (filled: contact, empty: no contact). (control: internode: $n = 32$ contacts from 16 animals, node: $n = 9$ contacts from 8 animals; rem: $n = 6$ contacts from 6 animals). **B**, **C** Dynamics of microglial processes contacting an internode vs a node in myelinated slices (internode: $n = 32$ contacts from 16 animals, node: $n = 9$ contacts from 8 animals). **D** Trajectories (with t0 position as reference, distance in μm) of microglial process tips in myelinated slices, initially contacting a node (nodal contact) or not (wo contact). wo contact: $n = 14$, nodal contact: $n = 7$, from 7 color coded animals. Type II Wald $\chi^2$ (two-sided analysis), $p = 1.679e-8$ (for quantification, see Fig. S4F). **E** Instantaneous process velocity in myelinated slices. wo contact: 280 measures from 14 trajectories in 7 animals, nodal contact: 140 measures from 7 trajectories in 7 color coded animals. **F**, **G** Dynamics of microglial processes contacting a node in control (node ctrl) or remyelinating slices (node rem). ctrl: $n = 9$ contacts from 8 animals, rem: $n = 6$ contacts from 6 animals. **H** Trajectories (with t0 position as a reference, distance in μm) of microglial process tips in remyelinating slices, initially contacting a node or not. wo contact: $n = 12$, nodal contact: $n = 6$, in 6 color coded animals. Type II Wald $\chi^2$ test (two-sided analysis), $p = 2.2e-16$ (for quantification, see Fig. S4H). **I** Instantaneous process velocity in remyelinating slices. wo contact: 240 measures from 12 trajectories in 6 animals, nodal contact: 120 measures from 6 trajectories in 6 color coded animals. Scale bars: **A** 10 μm. **B**, **C**, **F**, **G** Two-sided Mann–Whitney test, **E**, **I** Type II Wald $\chi^2$ test (two-sided analysis). *$P < 0.05$, **$P < 0.01$, ***$P < 0.001$, ****$P < 0.0001$, ns not significant; bars and error bars represent the mean ± s.e.m. For detailed statistics, see Supplementary Table.

strengthened in the remyelinating context. We thus questioned which signal(s) could be regulating this interaction.

**CX3CR1, P2Y12, and P2Y13 are not required for microglia-node interaction in physiological condition.** Various signaling pathways mediating neuron-microglia crosstalk have been uncovered in physiological and pathological contexts[38]. Among them, we first focused on CX3CR1 pathway, taking advantage of the fact that CX3CR1-GFP mouse strain corresponds to a knock-in of *GFP* into the *CX3CR1* gene locus, leading to a loss of function of this gene in homozygous mice. By comparing

CX3CR1$^{GFP/GFP}$ with CX3CR1$^{GFP/+}$ littermates in myelinated cultured slices, we found no significant difference in the percentage of nodes contacted ($25.1 \pm 2.5\%$ vs $20.8 \pm 2.6\%$ respectively, Fig. S5A–B). We confirmed these results in vivo, where no significant difference was observed in the mean percentage of contacted nodes between CX3CR1$^{GFP/GFP}$ and control mice (Fig. S5C–D).

The microglial P2Y12 purinergic receptor has also been involved in multiple microglia-neuron interactions, and is highly expressed by microglia in myelinated cerebellar slices ex vivo (Fig. S6A). In order to address whether this pathway was required for microglia-node

contact, we first inhibited the P2Y12R receptor specifically using PSB0739 (1 μM), a highly potent P2Y12 antagonist, and found that it is not required for microglia-node interaction (15.1 ± 1.2% of nodes contacted in control vs 15.5 ± 1.7% following treatment, Fig. S6B–C). To further assess the potential role of the P2Y receptor family and exclude any compensatory mechanisms, we also inhibited P2Y13R, which participates in microglia dynamics control[39], concomitantly to P2Y12R by using the MRS2211 inhibitor (50 μM). Again, there was no significant variation in the number of nodes contacted in control vs treated slices (15.1 ± 1.5% vs 14.3 ± 1.3% respectively, Fig. S6D–E), confirming that P2Y12R and P2Y13R are not required for microglia-node interactions in physiological condition.

**Neuronal electrical activity modulates microglia-node interaction**. As reciprocal interactions between electrically active neurons and microglia have been established, we then addressed whether changes in neuronal activity might impact microglia-node contacts. Purkinje cells are spontaneously active in organotypic cerebellar slices, as assessed by loose-cell attached recordings ($n = 13$ cells from 9 animals, Fig. S7A). To modulate neuronal activity, we used two pharmacological agents, Apamin (Apa, 500 nM) and Tetrodotoxin (TTX, 500 nM). Apamin is a blocker of SK2 channels, which leads to a specific increase of Purkinje cells firing in the cerebellum[40], while TTX is a well described blocker of action potential generation through voltage-gated sodium channel inhibition[41]. Apamin treatment lead to a 60% increase of Purkinje cell instantaneous firing frequency (Ctrl: 16.09 ± 3,47 Hz; Apamin: 26.12 ± 7,14 Hz, Fig. S7B–C), whereas TTX completely blocked their activity (Fig. S7A–B). We first confirmed that both TTX and Apamin do not affect global morphology and dynamics of microglial cells (Fig. S8). We then addressed whether microglia-node contacts were modified following these pharmacological treatments. Following 1-h TTX treatment of myelinated slices, microglia-node contacts were significantly decreased by 20% (21.3 ± 2.4% nodes contacted in control vs 16.9 ± 2.0% in treated slices, Fig. 5A–C), while a 1-h Apamin treatment on myelinated slices significantly increased by 20% the percentage of nodes contacted (16.9 ± 1.7% nodes contacted in control vs 20.4 ± 1.2% in treated slices, Fig. 5D–F).

Taken together, these data demonstrate that microglial contacts at nodes are modulated by neuronal activity, adding further insight into the role of microglia as a neuronal activity sensor.

**Microglia-node interaction is modulated by potassium release at nodes**. Recent works suggest that microglia dynamics can be modulated by extracellular potassium concentration[10,42]. To assess the potential involvement of nodal potassium flux in microglia-node interaction, we treated myelinated cultured slices with the large spectrum inhibitor of potassium channels tetraethylammonium (TEA, 30 mM, Fig. 6A) for 1 h. We first confirmed that TEA does not affect global morphology and dynamics of microglial cells (Fig. S8). This treatment however resulted in a 40% decrease of microglia-node interaction, with 11.4 ± 1.1% of the nodes contacted in the treated slices, compared to 18.8 ± 1.1% in control condition (Fig. 6B, C). High concentration of TEA is expected to decrease Purkinje cells activity, which would lead to reduced interaction. We thus compared the normalized effects of TEA and TTX and observed that the reduction of microglia-node interaction is significantly more pronounced after TEA treatment than after TTX treatment (40 and 20% decrease respectively), suggesting that the outward potassium current existing at axonal resting potential may also participate in modulating microglia-node interaction.

We then addressed how TEA impacts the dynamics of microglial process in myelinated slices, by comparing processes contacting a node or an internode (Fig. 6D–I and Movie 8). Contrary to the preferential interaction of microglial processes at nodes found in untreated slice cultures (Fig. 4), in the TEA treated slices, the dynamics of contact was similar between processes with internodal or nodal contact (49.7 ± 4.9 % of timeframes with contact at internodes vs 55.2 ± 9.3 % at nodes, Fig. 6E) and consecutive number of timeframes with contact was comparable (internode: 8.5 ± 0.9 frames; node: 10.0 ± 1.8 frames; Fig. 6F). Similarly, in myelinated slices treated with TEA, the process trajectory and velocity are not significantly different for processes with and without nodal contacts (Fig. 6G–I).

The two-pore domain channel THIK-1 was recently identified as the main $K^+$ channel expressed in microglia[10]. We first showed that THIK-1 mRNA is expressed by microglial cells in myelinated and remyelinating condition in our slice model (Fig. S9A, C respectively), and that it is not expressed by Purkinje cells (Fig. S9B, D). We detected THIK-1 mRNA expression in the majority of microglial cells, with 76.4 ± 3.2% of microglial cells clearly expressing this marker in myelinated tissue and 66.5 ± 8.0% in remyelinating condition (Fig. S9E, F, $n = 3$ animals per condition, $n = 200$ and $n = 209$ total cells analyzed respectively). To confirm the role of $K^+$ flux in microglia-node interaction, we treated myelinated cultured slices for 1 h with tetrapentylammonium (TPA, 50 μM), a blocker of THIK-1, (Fig. 6J). We observed a significant 36% decrease of microglia-node interaction, with 10.8 ± 1.2% of the nodes contacted in the treated slices, compared to 17.1 ± 1.4% in control condition (Fig. 6K–L), an effect which is similar in amplitude to the effect of TEA treatment.

Taken together, these results confirm that microglia preferential interaction with nodes of Ranvier depends on potassium level within the nodal area, with a key role of microglial THIK-1 $K^+$ channel.

**The alteration of microglia-node interaction by $K^+$ channel inhibitors impairs the microglial switch towards a pro-regenerative phenotype and decreases remyelination**. Having confirmed that node-microglia communication also depends on $K^+$ flux in remyelination (Fig. 7A–C), we next addressed whether altering microglia-node interaction at remyelination onset could affect the repair process in LPC-demyelinated cultured cerebellar slices. To address this question, we thus treated cerebellar slices with either TEA or TPA at the onset of remyelination and studied whether this could affect the microglial switch from pro-inflammatory towards pro-regenerative phenotype, by quantifying the percentage of microglial cells expressing the pro-regenerative marker IGF1 (Fig. 7D, E, TEA and Fig. 7H, I, TPA) and the pro-inflammatory marker iNOS (Fig. S9G, H, TEA and Fig. S9I, J, TPA). We observed a significant 22% decrease in IGF1 population following TEA treatment (ctrl: 74.2 ± 2.8%; TEA: 57.5 ± 1.9%; Fig. 7E), and a significant 25% decrease following TPA treatment (ctrl: 79.7 ± 2.2%; TPA: 60.0 ± 1.0%; Fig. 7I). Conversely, we observed a significant 21% increase in iNOS+ population following TEA treatment (ctrl: 57.2 ± 0.9%; TEA: 69.5 ± 1.7%; Fig. S9H), and a significant 26% increase following TPA treatment (ctrl: 60.0 ± 1.0%; TPA: 75.9 ± 2.2%; Fig. S9J). Microglia-node $K^+$ signaling dysregulation at onset of remyelination thus results in altered microglial phenotypic switch.

We next quantified the rate of remyelination of Purkinje axons in demyelinated slices treated with TEA or TPA, compared to untreated slices (Fig. 7F–G, J–K). We observed a significant 24% decrease of the remyelination rate following TEA treatment (ctrl:

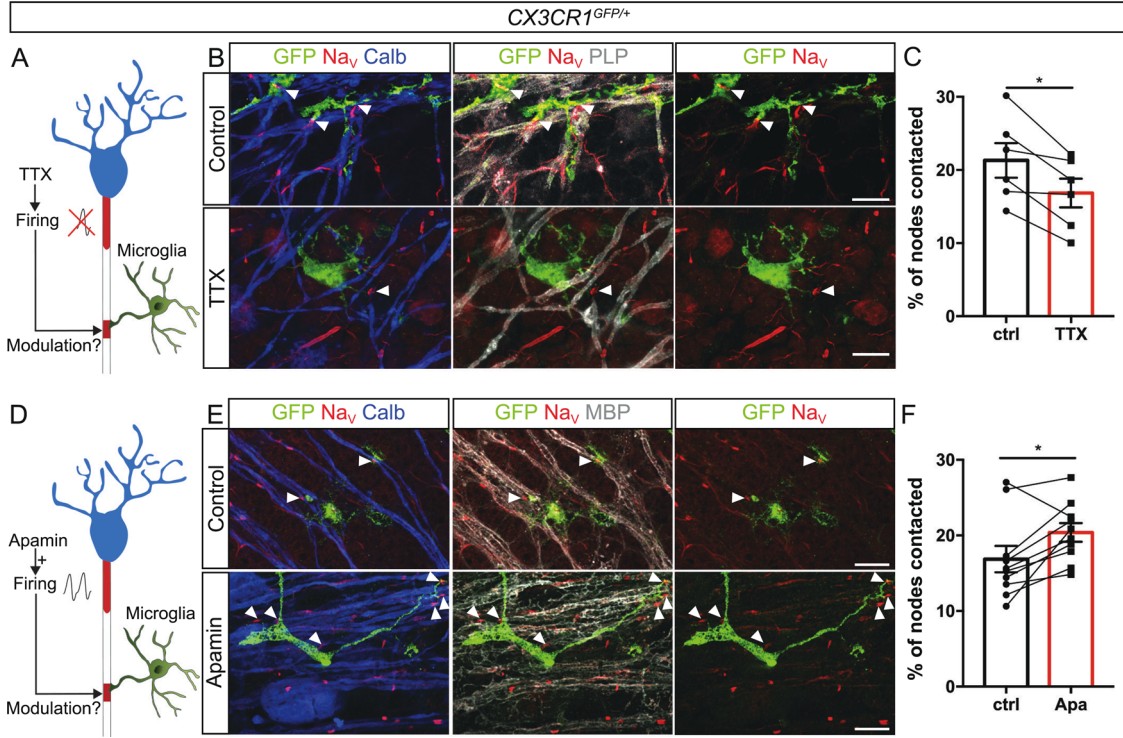

**Fig. 5 Neuronal activity modulates microglia-node of Ranvier interaction. A, D** Experimental designs. **B, C** In myelinated CX3CR1$^{GFP/+}$ cerebellar organotypic slices, microglia (GFP, green) contacts with nodes (Na$_V$, red) are reduced following electrical activity inhibition with tetrodotoxin (TTX). **E, F** They are conversely increased following electrical activity activation using apamin (Apa). Arrowheads show the nodes of Ranvier contacted by microglia. **C, F** Percentage of nodes of Ranvier contacted by microglial cells in control (ctrl) vs 1-h treated slices from the same animal (**B–C**) TTX (500 nM, $n = 6$ animals), **E, F** Apamin (500 nM, $n = 10$ animals). Scale bars: **B, E** 10 μm. **C** Two-sided Paired t-test; **F** Two-sided Wilcoxon matched pairs test. *$P < 0.05$, **$P < 0.01$, ***$P < 0.001$, ****$P < 0.0001$, ns not significant; bars and error bars represent the mean ± s.e.m. For detailed statistics, see Supplementary Table.

47.7 ± 2.6%; TEA: 36.4 ± 2.5%; Fig. 7G), and a significant 23% decrease following TPA treatment (ctrl: 37.5 ± 2.7%; TPA: 28.9 ± 1.7%; Fig. 7K).

To confirm these observations in vivo, we induced demyelination in mouse dorsal spinal cord by LPC focal injection and placed at 9 DPI a pump allowing the delivery of TPA (50 μM) or its carrier solution directly above the lesion (Fig. 8A). The animals were perfused 2 days later and the microglial cell phenotype (using IGF1 stainings) and the remyelination rate were assessed (Fig. 8B–E). We observed a significant 15% decrease of IGF1$^+$ microglia (Fig. 8B, C) and a significant 2.6 fold increase of the non remyelinated area (Fig. 8D, E) within the lesions in TPA treated animals compared to controls.

These results are further supported by our observation that, in remyelinating mouse dorsal spinal cord lesion following LPC-induced focal demyelination, microglia contacting nodal structures have a more pro-regenerative phenotype, with a 31% increase of IGF1$^+$ pro-regenerative cells and a 35% decrease of iNOS$^+$ pro-inflammatory cells in node-contacting microglial population compared to non-contacting cells (Fig. S9K–M).

Taken together these data demonstrate that nodal K$^+$ modulates the remyelination process, by promoting the pro-regenerative phenotype of microglia after myelin injury. This effect is associated and likely mediated by changes in microglia-node of Ranvier interaction.

## Discussion

Here we show that microglia establish direct contacts at nodes of Ranvier both in mouse and human tissue. These microglial contacts co-exist on a given node with contacts from astrocytes and OPCs processes, establishing the node as a privileged site for axons to interact with their glial environment. Using in vivo and ex vivo mouse models, we show that this interaction is stable in healthy tissue but also during remyelination, with an increased percentage of nodes contacted in this latter situation. In addition, we provide evidence that microglial process contacting nodes of Ranvier have reduced dynamics compared to processes moving freely or contacting internodal areas, reinforcing the hypothesis of a specific interaction rather than a random "surveilling" contact between microglia and nodes of Ranvier. This preferential interaction at the nodes of Ranvier depends on neuron activity and potassium ion release. When altering this interaction during remyelination by disrupting nodal K$^+$ flux or blocking microglial K$^+$ channels, microglial cells acquire a more pro-inflammatory phenotype, which associates with reduced remyelination (model Fig. 9).

**Microglial contact at nodes is stable and depends on neuronal activity and physiological status of the tissue.** We show that the nodes of Ranvier are specific sites for neuron-microglia interaction, this interaction likely corresponding to microglial sensing of neuronal activity, mediated through specific signaling. We first observed that the majority of microglial cells interact with nodes of Ranvier, regardless of the CNS area considered, with about a fifth to a fourth of the nodes contacted in healthy condition in spinal cord and cerebellar mouse tissue. The increased rate of contact in remyelinating condition could be linked to neuronal transient hyperactivity[43]. Whether frequency of nodal contacts depends on the neuronal subtype or activity pattern is unknown.

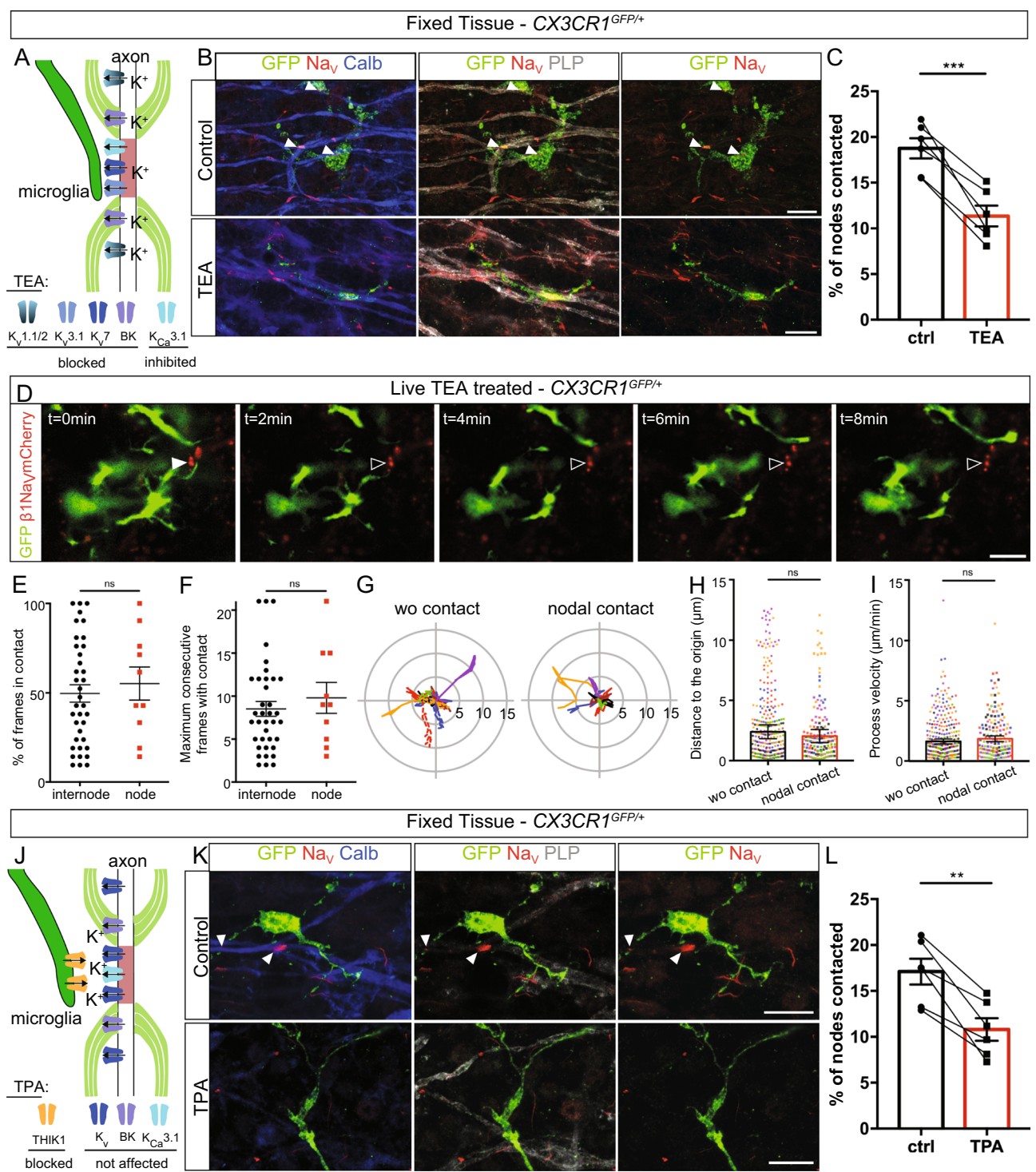

Using live-imaging we were able to demonstrate for the fist time the stability of these microglia-nodes contacts, strongly arguing for a specific interaction rather than a random survey of the local environment by microglial processes. Whether the reduced interaction stability in perilesional tissue relates to altered axonal physiology or increased local inflammatory signals remains uncertain.

When addressing the mechanisms supporting these microglia-nodes interactions, we demonstrate that neuronal electrical activity and nodal efflux of potassium are playing a key role, by showing that pharmacological inhibition of nodal and perinodal potassium channels leads to a 40% decrease of the contact, and

that blocking THIK-1, the main microglial potassium channel linked with $K^+$ homeostasis[10] reproduces this effect. Various potassium channels have been suggested to be at play at the node, depending on the neuronal subtype considered, with Kca3.1 being present at nodes of Ranvier in Purkinje cells[44] and TRAAK and TREK channels recently described as playing a major role at PNS nodes, and in several CNS areas, such as the cortex, hippocampus and spinal cord[45,46]. Thus, the microglia-node interaction may be modulated through axonal potassium efflux mediated by various players depending of the neuron considered, with the microglial THIK-1 channel allowing the read-out of this signal.

**Fig. 6 Microglia preferential contact with nodes depends on potassium fluxes. A** Experimental design. **B** In myelinated CX3CR1$^{GFP/+}$ cerebellar organotypic slices, microglia (GFP, green) contacts with nodes (Na$_V$, red) are reduced following potassium channel inhibition by tetraethylammonium (TEA). Arrowheads indicate the nodes of Ranvier contacted. **C** Percentage of nodes of Ranvier contacted in control condition or following 1-h TEA treatment (**B, C**, 30 mM, $n = 6$ animals); the mean values per animal are individually plotted. **D** Microglial cell (green) initially contacting a node (red) in a myelinated slice treated with TEA. Arrowheads show the initial contact position (filled: contact, empty: no contact). **E, F** Dynamics of microglial tips contacting an internode vs a node in myelinated slices treated with TEA (**D–F**, internode: $n = 37$ contacts from 12 animals, node: $n = 10$ contacts from 8 animals). **G** Trajectories (with t0 position as reference, distance in µm) of microglial process tips whether they were initially contacting a node (nodal contact) or not (wo contact) at t0 (wo contact: $n = 14$; nodal contact: $n = 7$; from 7 color coded movies from 6 animals). **H** Distance between the process tip and its position at t0 for each timepoint in myelinated slices treated with TEA (wo contact: 280 measures from 14 trajectories in 6 animals, nodal contact: 140 measures from 7 trajectories from 7 color coded movies in 6 animals). **I** Instantaneous process velocity in myelinated slices (wo initial contact: 280 measures from 14 trajectories in 7 animals, initial nodal contact: 140 measures from 7 trajectories in 7 color coded animals). **J** Schematic of the experimental design. **K** Microglia (GFP, green) contacts at nodes (Na$_V$, red) are importanlty reduced following THIK-1 inhibition by tetrapentylammonium (TPA). **L** Percentage of nodes of Ranvier contacted by microglial cells in control condition or following 1-h TPA treatment (**K, L**, 50 µM, $n = 6$ animals); the mean values per animal are individually plotted. Scale bars: (**B, D, K**) 10 µm. **C, L** Two-sided Paired t-test; **E, F** Two-sided Mann–Whitney test; **H, I** Type II Wald $\chi^2$ test (two-sided analysis). *$P < 0.05$, **$P < 0.01$, ***$P < 0.001$, ****$P < 0.0001$, ns not significant; bars and error bars represent the mean ± s.e.m. For detailed statistics, see Supplementary Table.

Although some molecular mechanisms underlying microglia-node contact remain to be further deciphered, our results highlight a novel way of axo-glial interaction.

**The node of Ranvier: a neuron-glia communication hub?** It has previously been described that OPCs and perinodal astrocytes contact nodes of Ranvier[32], and that these cells might interact at a given node[47]. We now show that microglia can also form multipartite contacts with these other glial cell types, which suggests that nodes of Ranvier may act as a hub of intercellular communication in the CNS. This is further supported by our 3D reconstruction data showing that microglia establish a physical contact both with the nodal constituents and the other contacting glial cell.

This nodal "communication hub" could in particular participate in transmitting information on neuronal function and neuronal health to the glial environment and glia could in return modulate neuronal physiology. Oligodendroglial lineage cells have been shown to sense neuronal activity, which modulates their proliferation, differentiation and the myelination process[41,48,49]. Furthermore, electrophysiological paired recordings show that OPCs adjacent to neurons can sense their discharge, and extracellular stimulation induces slow K$^+$ currents in OPCs, showing they could sense K$^+$ variations linked with activity[50]. This suggests that both OPCs and microglia could have a read-out of neuronal activity at nodes through similar signaling pathways. Nodal protein clusters could further guide myelination initiation along some axons both in development and repair[51,52].

In addition to this node-OPC dialog, astrocyte-node interaction has been implicated in buffering K$^+$ at the node[53], but also recently in the regulation of axonal conduction velocity, by modulating nodal gap length and myelin thickness. This later effect relies on the secretion of Serpine2[54], a thrombin inhibitor also expressed by microglial cells in mice[19,55]. This neuronal crosstalk with multiple glial cells at node could be of importance for network synchronization and adaptive plasticity in healthy condition, as well as adequate repair in disease[49].

**Microglia-node of Ranvier interaction in disease: damaging or protective?** Activated microglia are known to have a dual role in disease, with a deleterious impact in case their pro-inflammatory phenotype is maintained inappropriately, and an active role in repair, in particular when they acquire a pro-regenerative phenotype[14,21]. Activated microglia have been observed in normal appearing white matter (NAWM) in MS tissue and MS models, together with nodal area alteration[56–58]. In addition,

some nodes of Ranvier are contacted by macrophages even prior demyelination and it has been hypothesized that they might be an early site of axonal damage[57]. It appears however that the cells actively attacking nodal domains are rather monocyte derived macrophages than microglial cells[59]. Our observation that microglial-node contact is destabilized in perilesional area at the peak of demyelination is compatible with this observation.

**An increased microglia-node interaction might favor remyelination.** In both our in vivo and ex vivo models, we report that microglia contact nodal structures, including heminodes and node-like clusters at the onset of remyelination. Strikingly, as observed in our ex vivo model, the microglial cell processes at nodes are less motile during ongoing remyelination compared to control myelinated tissue. This reinforced attraction/stability could be linked to increased potassium release at the sites where nodal structures reaggregate or to increased microglial sensitivity to this K$^+$ signal in a remyelinating condition. Regarding K$^+$ flux in remyelination, it is described that a transient neuronal hyperactivity can be observed in the first phasis of remyelination[44], suggesting K$^+$ efflux might be increased at that time. Furthermore, this pathway and other(s) could synergize, such as P2Y12R, which is known to potentiate THIK-1[10].

Nodes of Ranvier disruption and reassembly following demyelination have a major impact on neuronal physiology. In addition, several lines of evidence have shown that neuronal activity affects the (re)myelination process[41,60,61]. Early nodal reclustering along axons might lead to restored saltatory-like (or "microsaltatory") axonal conduction[32] and generate a local increase of K$^+$ efflux. This local modulation might trigger microglial behavior changes and participate in the regulation of microglial activation state by neuronal activity[62,63], including the pro-inflammatory to pro-repair switch observed concomitantly to the onset of remyelination[14,64,65]. Intracellular K$^+$ decrease has been associated with microglial inflammasome expression[10,66]. The axonal K$^+$ locally released at nodal structures when they reassemble could thus contribute to modulate microglial K$^+$ flux and subsequently limit inflammasome expression in the microglial cells with nodal contacts. Conversely, the restoration of stable microglial contacts at nodal structures could allow for microglia to buffer reassembled nodal area and modulate activity at nodes, as neurons may be transiently hyperexcitable at onset of remyelination[43,67].

To address the functional role of microglia-node interaction in repair, we took advantage of the modulation of the microglia-node interaction by potassium flux. We first inhibited potassium efflux at onset of remyelination and further targeted THIK-1, the

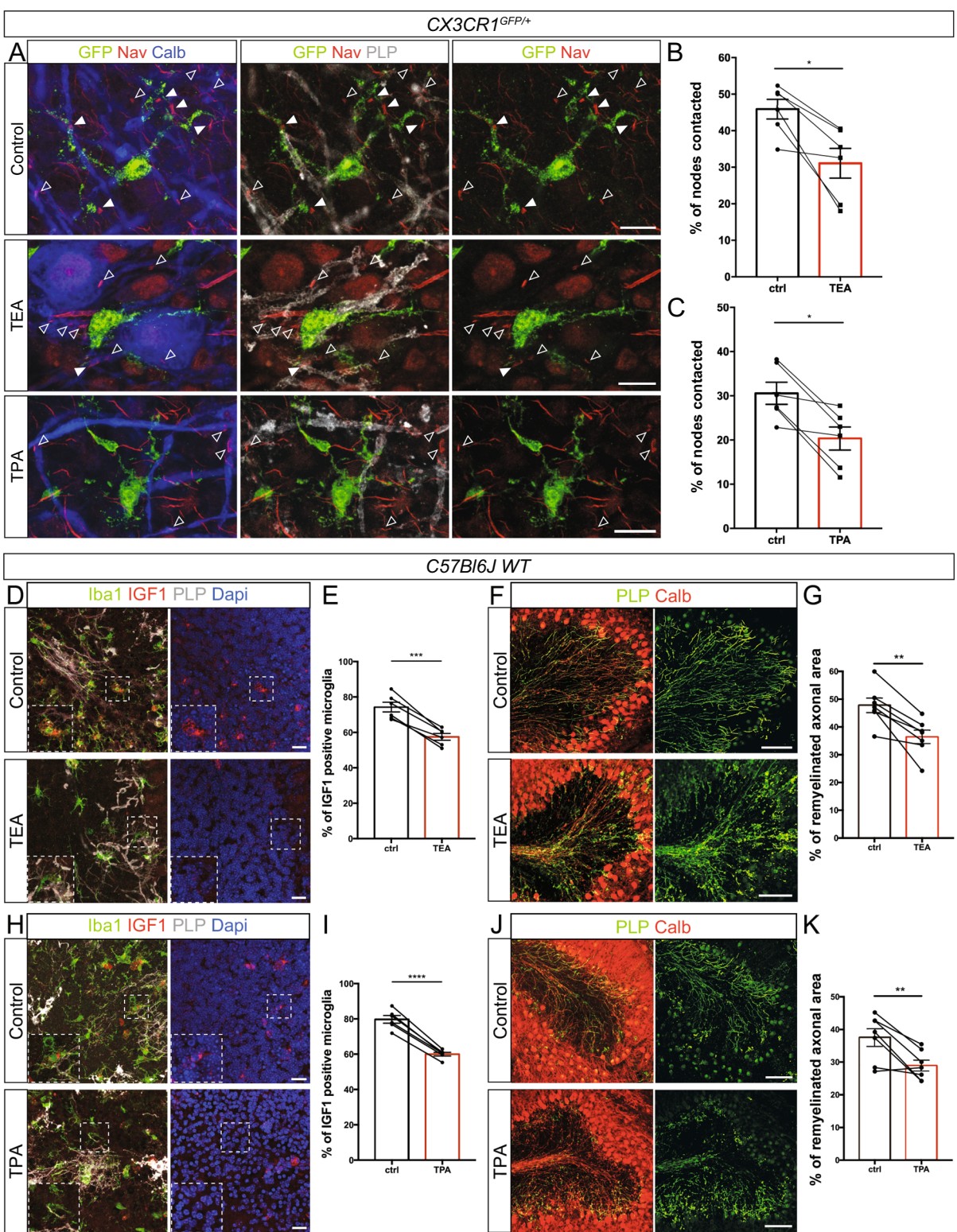

main channel implicated in microglial K$^+$ conductance[10]. Interestingly, THIK-1 channel is expressed in microglial cells in the CNS in healthy as well as MS tissues[55,68]. These different pharmacological approaches affect the microglial switch towards a pro-regenerative profile, associated with decreased remyelination. This suggests that neuron dialog with microglia through K$^+$ signaling might influence myelin regeneration capacity.

Microglia contact at nodal structures could allow for a read-out of the neuron physiological status and orientate microglial phenotype, which in turn could modulate neuronal survival and remyelination. This reciprocal interaction might be of importance in different pathological contexts in the CNS.

Recent data suggest that the impact of neuronal activity on OPCs recruited at the lesion follows a two-step mechanism, with consecutive time windows, first facilitating proliferation of OPCs, then promoting OLs differentiation and myelination[43,60,61,69]. This parallels the biphasic activation of microglial cells following demyelination, with a switch from the pro-inflammatory to the

**Fig. 7 Altered K$^+$ fluxes following demyelination leads to a reduced number of pro-regenerative microglia and impairs remyelination ex vivo. A** In remyelinating CX3CR1$^{GFP/+}$ cerebellar organotypic slices, microglial (GFP, green) contacts with nodes (Na$_V$, red) are importantly reduced following potassium channels inhibition by TEA and TPA treatments. Arrowheads show the nodal structures (filled: contacted by microglia; empty: not contacted). **B, C** Percentage of nodal structures contacted by microglial cells in control remyelinating condition or following 1-h TEA (**A, B**, 30 mM, $n = 6$ animals) or TPA treatment (**A, C**, 50 μM, $n = 6$ animals); the mean values per animal are shown as dots. **D–H** In remyelinating C57bl6/J cerebellar organotypic slices, the number of microglial cells expressing IGF1 is decreased following potassium channel inhibition by TEA (**D**) or THIK-1 inhibition by TPA treatment (**H**). **E–I** Percentage of IGF1$^+$ microglial cells at remyelination onset, with no treatment (ctrl) or with TEA (**D, E**, 2 h, 30 mM, $n = 6$ animals) or TPA treatment (**H, I**, 2 h, 50 μM, $p = 2.195e-5$, $n = 6$ animals), the mean values per animal are shown as dots and paired with the corresponding control. **F, J** In remyelinating C57bl6/J cerebellar organotypic slices, remyelination is reduced following TEA (**F**) or TPA (**J**) treatment. **G, K** Percentage of axonal area remyelinated in LPC-demyelinated slices without (ctrl) or with TEA (**F, G**, 2 h, 30 mM, $n = 7$ animals) or TPA treatment (**J, K**, 2 h, 50 μM, $n = 7$ animals). Scale bars: **A** 10 μm; **D, H** 20 μm; **F, J** 100 μm. **B, C, E, G, I, K** Two-sided Paired t-test. *$P < 0.05$, **$P < 0.01$, ***$P < 0.001$, ****$P < 0.0001$, ns not significant; bars and error bars represent the mean ± s.e.m. For detailed statistics, see Supplementary Table.

pro-regenerative state, with recently published data suggesting that both stages are required for optimal repair[14,16,65]. Our results show that the recovery of a stable microglia-node interaction at onset of remyelination could influence the microglial phenotypic switch and contribute to accumulation of pro-regenerative microglia. Neuronal activity could then play a role in remyelination both through direct OL lineage regulation and indirectly through modulating microglial phenotypic orientation.

The preferential interaction of the main glial cell types at nodes of Ranvier forming a "neuro-glial communication hub" may thus contribute to efficient repair orchestration.

## Methods

**Animal care and use.** The care and use of mice conformed to institutional policies and guidelines (UPMC, INSERM, French and European Community Council Directive 86/609/EEC). The following mouse strains were used: C57bl6/J (Janvier Labs), CX3CR1-GFP (gift from Prof S. Jung, Weizmann Institute of Science, Israel[70]) and Thy1-Nfasc186mCherry (gift from Prof P.J. Brophy, University of Edinburgh, UK).

**Focal demyelination of mouse spinal cord.** Following intraperitoneal injection of Ketamin/Xylazin in NaCl 9‰ (respectively 110/25 mg/kg, Centravet), a small incision was made between thoracic and lumbar vertebrae to access the spine and inject with a glass capillary 1 μl of lysophosphatidylcholin diluted into NaCl 9‰ (LPC, 10 mg/ml; Sigma-Aldrich) or NaCl 9‰ for the sham condition. Following surgery, mice were stitched and placed into warming chambers (Vet tech solution LTD, HE011). At either 7-day post-injection (7 DPI peak of demyelination) or 10–11 DPI (early remyelination phase), the mice were euthanized with Euthasol (Centravet) and transcardially perfused with 2% PFA (Electron Microscopy Services).

**Osmotic pump surgery.** The surgery to place the micro-osmotic pump (Alzet, 1003D) above the demyelinating spinal cord lesion was made 9 days post LPC injection. The day before surgery, the pumps were pre-filled with NaCl 9‰ (Ctrl) or TPA diluted in NaCl 9‰ (TPA, 50 μM). The pumps were subsequently connected to a cannula with three depth-adjustment spacers (Alzet, brain infusion kit III) and incubated overnight at 37 °C in sterile saline solution. Prior to surgery, the mice received an intraperitoneal injection of Ketamin/Xylazin in NaCl 9‰ (respectively 110/25 mg/kg, Centravet) and a small incision was made in the skin above the site of LPC injection. The pre-filled pump connected to the cannula was inserted subcutaneously and the spinal cord meninges were carefully removed above the lesion. The extremity of the cannula connected to the pump was placed above the lesion without touching the surface of the spinal cord and fixed with glue. The skin incision was stitched and the animals were monitored closely 24 h post-surgery. After 2 days, the animals were intracardially perfused with 4% PFA and the spinal cord was collected through a ventral laminectomy. The tissue was then processed as described in the Tissue Preparation section.

**Spinal glass window surgery.** Surgery protocol was adapted from Fenrich et al.[34]. Briefly, following an intraperitoneal injection of Ketamin/Xylazin as described above, the dorsal skin and muscles were incised, and the spine immobilized with two spinal forks placed at T12 and L2 vertebrae, with Tronothane 1% (Lisa-Pharm) applied at the point of junction with the forks. Two staples were then fixed along the transverse processes of the vertebrae as a support for a reshaped paperclip, stabilized with glue (Cyanolite) and dental cement (Unifast Trad 250 mg/250 mg, GC Dental Products Corp). A dorsal laminectomy was performed using a high-speed drill with a carbide bur. Spinal cord was then hydrated with a solution PBS/penicillin-streptomycin/Naquadem (Dexaméthasone 0.05%, Intervet) and LPC (or

NaCl) was injected as described above. A glass window cut from a glass coverslip (Menzel-Glaser, 0.13–0.16 mm) was cleaned, dried and placed above silicon (Kwik-Sil, World Precision Instruments) directly applied on the spinal cord. The window was fixed with glue and dental cement. Finally, the animal received a subcutaneous injection of Buprenorphine (0.1 mg/kg, Centravet) and was placed in a warming chamber.

**Cerebellum organotypic slice culture.** Ex vivo culture protocol was adapted from[36,37]. Briefly, P8 to P10 mouse cerebella were dissected in ice cold Gey's balanced salt solution complemented with 4.5 mg/ml D-Glucose and penicillin-streptomycin (100 IU/mL, Thermo Fisher Scientific). They were cut into 250 μm parasagittal slices using a McIlwain tissue chopper and the slices placed on Millicell membrane (3–4 slices per membrane, 2 membranes per animal, 0.4 μm Millicell, Merck Millipore) in 50% BME (Thermo Fisher Scientific), 25% Earle's Balanced Salt Solution (Sigma), 25% heat-inactivated horse serum (Thermo Fisher Scientific), supplemented with GlutaMax (2 mM, Thermo Fisher Scientific), penicillin–streptomycin (100 IU/mL, Thermo Fisher Scientific), and D-Glucose (4.5 mg/ml; Sigma). Cultures were maintained at 37 °C under 5% CO$_2$ and medium changed every two to three days. Experiments were analyzed at 4 days in vitro (DIV) for myelinating condition and at 10 to 11 DIV for myelinated control and remyelinating conditions.

**Lentiviral transduction of cerebellar slices.** For live-imaging ex vivo, nodes of Ranvier were detected with β1Na$_v$mCherry expressed under the control of the Synapsin promoter. Briefly, the pEntr-β1Na$_v$mCherry plasmid was recombined with the pDestSynAS (generated by Philippe Ravassard, ICM), using the Gateway LR clonase kit from Thermo Fisher Scientific[37]; detailed description on demand) and the corresponding lentivirus produced by the ICM vectorology plateform. Transduction was performed immediately following slice generation by addition of the lentiviral solution directly onto the slices placed on Milicell membranes (1 μl/slice at a final concentration of ≈10$^9$ VP/μl).

**Ex vivo treatments.** As culture systems may lead to increased variability between animals, we always cultured the slices from a given animal on two membranes. One of these membrane was treated, while the second one was kept untreated. This allowed us to minimize the potential variability and to do paired experiments.

To induce demyelination, for each animal, the slices on one membrane were incubated overnight at 6 DIV in 0.5 mg/ml LPC added to fresh culture medium, while the other membrane was kept as control.

To study microglia-node interaction in myelinated tissue, for each animal, the myelinated slices (10–11 DIV) on one membrane were left untreated (control), while the ones on the other membrane were treated with tetrodotoxin (TTX, 1 h, 500 nM, Tocris) or apamin (1 h, 500 nM, Sigma-Aldrich) to inhibit or activate neuronal activity respectively. Similarly, PSB0739 (3 h, 1 μM, Tocris) or MRS2211 (3 h, 50 μM, Tocris) were used to inhibit the microglial purinergic receptors P2Y12R and P2Y12R/P2Y13R respectively. Tetraethylammonium (TEA, 1 h, 30 mM, Sigma) was used to neuronal inhibit potassium channels and tetrapentylammonium (TPA, 1 h, 50 μM, Sigma) to inhibit the microglial THIK-1 channel. To study microglia-node interaction in remyelination, for each animal, the two membranes were demyelinated using LPC, and one of the demyelinated membranes was treated at 10–11 DIV with TEA (1 h, 30 mM) or TPA (1 h, 50 μM), while the other membrane was kept untreated as control.

To evaluate the functional impact of the perturbation of microglia-node interaction, demyelinated slices were treated at the very onset of remyelination (9.5 DIV) with 30 mM TEA or 50 μM TPA for 2 h and fixed 2 h or 15 h post-treatment (together with untreated demyelinated slices as control), to evaluate microglial phenotype and remyelination rate, respectively.

**Electrophysiology.** Myelinated cultured slices (10–13 DIV) were transferred to a recording chamber and continuously superfused with oxygenated (95% O$_2$ and 5% CO$_2$) aCSF containing (in mM): 124 NaCl, 3 KCl, 1.25 NaH$_2$PO$_4$, 26 NaHCO$_3$, 1.3

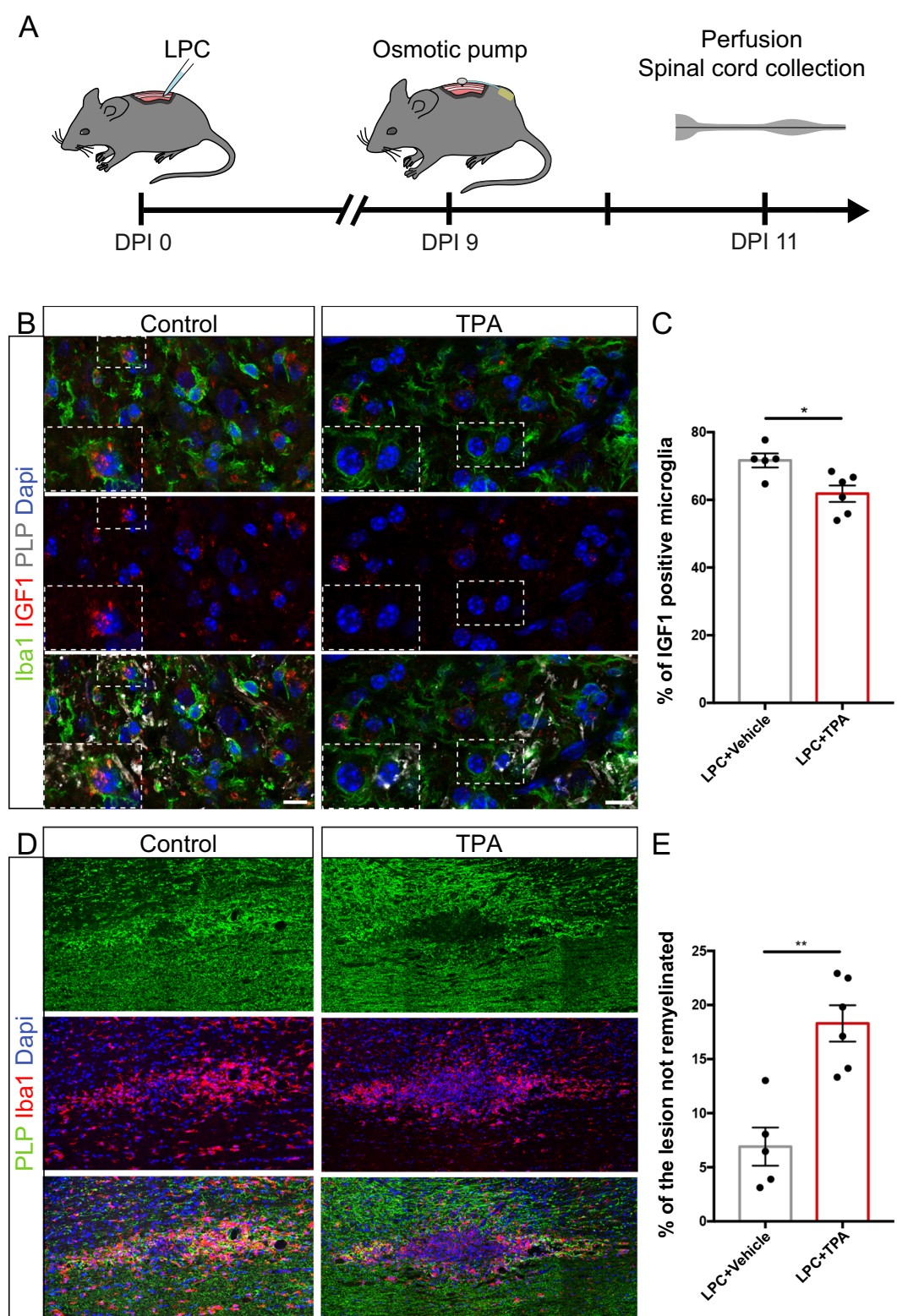

**Fig. 8 Altered K$^+$ flux read-out by microglia following demyelination leads to a reduced number of pro-regenerative microglia and impairs remyelination in vivo. A** Osmotic pumps were placed 9 days post LPC-injection to deliver NaCl 9‰ (Ctrl) or 50 μM TPA at the lesion site and the spinal cords were collected 2 days later. **B** In remyelinating dorsal spinal cord lesion, the number of microglial cells expressing IGF1 is decreased following TPA treatment. **C** Percentage of IGF1$^+$ microglial cells at 11 DPI in the remyelinating lesion, in control (Ctrl) or TPA condition (**B**–**C**, Ctrl: $n = 5$ animals, TPA: $n = 6$ animals). **D** Remyelination is reduced following TPA delivery at the lesion. **E** Percentage of the lesion devoided of myelin at 11 DPI, in control or TPA condition (**D**–**E**, Ctrl: $n = 5$ animals, TPA: $n = 6$ animals). Scale bars: **B** 10 μm, **D** 50 μm. **C**, **E** Two-sided Mann–Whitney tests. *$P < 0.05$, **$P < 0.01$, ***$P < 0.001$, ****$P < 0.0001$, ns not significant. Bars and error bars represent the mean ± s.e.m. For detailed statistics, see Supplementary Table.

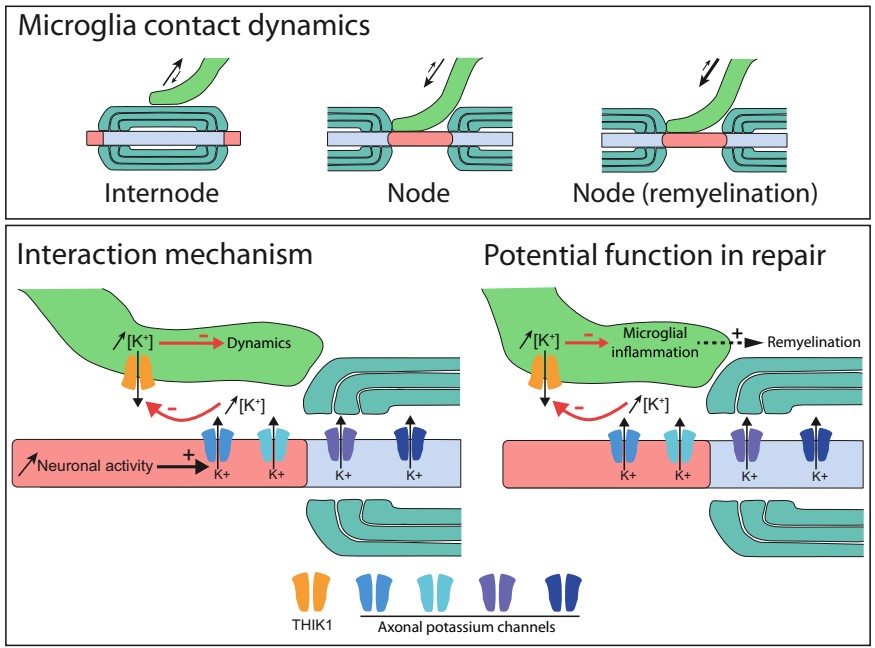

**Fig. 9 Model for microglial interaction at nodes of Ranvier.** Microglial contact with nodes of Ranvier depends on the tissular context. The interaction is in particular reduced in perilesional area at the peak of demyelination, and increased in remyelination compared to healthy myelinated tissue. Neuronal activity and nodal K+ efflux promote the interaction, with microglial read-out depending on THIK-1. Altering K+ efflux or THIK-1 activity in remyelination correlates with more pro-inflammatory microglia and decreased remyelination.

MgSO$_4$, 2.5 CaCl$_2$, and 15 glucose (pH 7.4). Purkinje cells were visualized under differential interference contrast optics using a 63X N.A 1 water immersion objective. Loose patch voltage clamp recordings of the spontaneous firing activity was performed at 32–34 °C with a borosilicate glass pipette filled with aCSF. Signals were amplified with a Multiclamp 700B amplifier (Molecular devices), sampled and filtered at 10 kHz with a Digidata 1550B (Molecular Devices). Data were acquired with the pClamp software (Molecular devices). To avoid any alteration of the spontaneous firing frequency of the cell by the patch procedure[71], the holding membrane potential was set to the value at which zero current is injected by the amplifier. The resistance of the seal ($R_{seal}$) was controlled and calculated every minute from the current response to a voltage step (50 ms; −10 mV). Only recordings with a $R_{seal}$ in the range of 10 to 200 MΩ and stable during the all recording procedure were included in the analysis. Spontaneous activity was tested in control condition and consecutively in the presence of apamin and TTX applied by bath perfusion. The firing rate was analyzed over a two minutes recording time window using a threshold crossing spike detection in Clampfit (Molecular devices) and calculated as the number of action potential current divided by the duration of the recording. The instantaneous frequency was reported since apamin treatment triggered clear bursts of the cells and calculated as the inverse of the interspike interval.

**Tissue preparation**

*Ex vivo cerebellar slices.* Cultured slices were fixed while attached to the membrane with either 4% or 1% PFA for 30 min at room temperature (RT) and washed with PBS.

*In vivo mouse brain and spinal cord.* For spinal cord preparation, the tissue was dissected following perfusion and post-fixed in PFA 2% for 30 min, washed in PBS and incubated in PBS, 15% sucrose for 3 days at 4 °C. Spinal cords were then embedded in OCT (Tissue-Tek, Sakura) or PBS, 15% Sucrose, 4% gelatin, and frozen in isopropanol onto carbonic ice.

For adult as well as P12 myelinating mouse brain tissue, animals were perfused with PFA 2% and the brain was dissected and post-fixed in PFA 2% for 30 min, washed in PBS, and incubated in PBS with successively 7, 15, and 30% sucrose for 3 days at 4 °C for cryoprotection. The tissues were then embedded in O.C.T (Tissue-Tek, Sakura).

Mouse spinal cords and brains were cut longitudinally and sagitally respectively. Human tissues were obtained after informed consent (healthy donors, UK MS Society Tissue Bank at Imperial College, London, under ethical approval by the National Research Ethics Committee 08/MRE09/31) as snap frozen blocks. Sections were cut using a cryostat Leica CM 1950 (30 and 15 μm thick for mouse and human respectively), collected on Superfrost + slides and stored at −80 °C until used.

*Electron microscopy.* Mice were transcardially perfused with PB 0.2 M, 4% PFA, and 0.5% glutaraldehyde (Electron Microscopy Services). Following 30 min post-fixation, the cerebellum was embedded in PB (0.2 M), 4% agarose (Sigma). Sagittal sections (50 μm thick) were generated using a VT1200S vibratome (Leica). The sections were then incubated in Ethanol 50% for 30 min, washed in PBS, and incubated with anti-Iba1 antibody (1:1000; Wako), in PBS, 3% NGS, 0.05% Triton-X, for 48 h on a rotating apparatus at room temperature (RT). The tissue was then washed in PBS and incubated for 1 h in biotinylated anti-rabbit IgG (1:200; Vector Labs, Burlingame, CA, USA), rotating at RT, further washed in PBS and incubated for 1 h in ABC (Vector Labs) solution, under rotation at RT. Following incubation, sections were washed 20 min with PBS and developed by incubating in 0.025% diamino-benzidine (DAB) and 0.002% hydrogen peroxide, in PBS. Sections were then processed for electron microscopy. Briefly, sections were post-fixed in 1% osmium tetroxide for 60 min, and dehydrated by ethanol and acetone immersion. A flat-embedding procedure was used, after which a semi-thin section was cut to find the area of interest and then ultrathin (60–80 nm) sections were cut with an ultramicrotome (Leica EM UC7, Wetzlar, Germany). The sections were stained with uranyl acetate and lead citrate to enhance contrast. Sections were examined with a Jeol Flash 1400 transmission electron microscope. Images of Iba1 immunolabeled microglial cells were captured with a Radius Morada digital camera. Once acquired, the brightness and contrast of the images were adjusted and structures of interest were highlighted using Photoshop software (Adobe, version CC).

**Antibodies.** Primary antibodies: mouse IgG2a anti-AnkyrinG (clone N106/36; 1:100), mouse IgG2b anti-AnkyrinG (clone N106/65; 1:75), and mouse IgG1 anti-Caspr (1:100), all from Neuromab; mouse IgG1 anti-Pan Na$_v$ (clone K58/35; 1:150; Sigma); mouse anti-Calbindin (1:500; Sigma), rabbit anti-Calbindin (1:300; Swant), rabbit anti-Caspr (1:300; Abcam), rat anti-PLP (1:10; kindly provided by Dr. K. Ikenaka, Okasaki, Japan), mouse IgG2b anti-MBP (1:200; SMI99, Sigma), rabbit IgG anti-Iba1 (1:500; Wako), chicken anti-GFAP (1:500; Aves Labs), rat anti-PDGFrα (1:100; BD Biosciences), rabbit IgG anti-TMEM119 (1:100; Sigma), rabbit IgG anti-P2Y12r (1:300; Alomone, human tissue), rabbit anti-P2Y12R (1:300; Anaspec, mouse tissue), chicken anti-GFP (1:250; Millipore), mouse IgG2a anti-iNOS (1:100; BD Biosciences), goat anti-IGF1 (1:50; R&D System), and chicken anti-mCherry (1:1000; Abcam). Secondary antibodies corresponded to goat or donkey anti-chicken, goat, mouse IgG2a, IgG2b, IgG1, rabbit and rat coupled to Alexa Fluor 488, 594, 647, or 405 from Invitrogen (1:500), or goat anti-mouse IgG1 DyLight from Jackson Immuno Research (1:600).

**Immunhistostainings**

*Immunofluorescent stainings.* Human sections were first fixed in paraformaldehyde (PFA) 4% for 5 min. For myelin protein staining, tissues were pre-incubated in absolute ethanol at −20 °C for 20 min. Tissues were blocked in PBS, 5–10%

Normal Goat Serum (50-062Z; Thermo Fisher Scientific), 0.2–0.4% Triton X-100 (Sigma), then incubated with primary antibodies in blocking solution overnight at room temperature, washed in PBS, and incubated with secondary antibodies for 1–3 h at RT in the dark. When applicable, tissues were immersed for 30 s in Hoechst solution (10 µg/ml, Euromedex). They were finally mounted under coverslip (VWR) with Fluoromount (Southern Biotech).

*Chromogenic immunohistochemistry on post-mortem brain tissues.* Snap frozen tissue blocks were rehydrated with PBS. Sections were then subjected to paraformaldehyde 4% for 10 min before elimination of endogenous peroxidase activity with 0.1% $H_2O_2$ (Sigma Aldrich) in PBS for 20 min. Blocking was performed using 5% normal sera before incubation with primary antibodies diluted in PBS containing 0.05% Triton 100-X and 5% sera. Binding of biotinylated secondary antibodies (Vector Laboratories) was visualized with the avidin-biotin horseradish peroxidase complex (Dako, Biotin Blocking System) followed by 3,3′-diaminobenzidine (DAB) (Vector Laboratories) as substrate. Primary antibody 2 was detected using the ABC-alkaline phosphatase detection system (Vector Laboratories), using Vector Blue as the substrate. Images were captured with a QICAM digital camera (QImaging Inc.).

**RNAscope mRNA detection.** RNAscope was performed on myelinated and remyelinating CX3CR1$^{GFP/+}$organotypic slices according to the supplier instructions, using RNAscope multiplex fluorescent detection reagents v2 kit (ACD, 323110), $H_2O_2$ and protease reagents (ACD, 322381) and kcnk13 (THIK-1), negative and positive control probes (respectively ACD, 535411, 320871, and 320881). Myelinated cultured slices (10 DIV) were fixed with 4% PFA for 30 min and washed with PBS. They were then treated 10 min with $H_2O_2$, transferred in target retrieval reagent for 10 min at 95 °C, followed by a 3 min treatment in ethanol. Slices were dried and treated with Protease III for 30 min, before being incubated with probes for 6 h at 40 °C. Amplification steps were subsequently performed at 40 °C using amplification solutions alternating with washes as follow: Amp1 for 30 min, Amp2 for 30 min and Amp3 for 15 min. The signal was then revealed using RNAscope multiplex FL v2 HRP-C1 signal for 15 min at 40 °C, followed by 30 min at 40 °C with a solution of Opal 650 (PerkinElmer, FP1496001KT, 1:1000) diluted in RNAscope multiplex TSA Buffer (ACD, 322809), and finally the reaction was stopped using RNAscope multiplex FL v2 blocker for 15 min at 40 °C. Immunofluorescence staining was then performed as described before to detect microglia and Purkinje cells using respectively anti-Iba1 (1:500; Wako) and anti-Calbindin (1:300; Swant) antibodies.

**Imaging**
*Confocal microscopy for ex vivo cultured slices.* Confocal microscopy was performed using an FV-1200 Upright Confocal Microscope and a Leica inverted SP8 with ×63 oil immersion objectives with 1.40 numerical aperture, using respectively metamorph and LasX software. For each acquisition, stacks of 1024 × 1024 pixel images (160.61 µm × 160.61 µm), including at least 10 Z-series with a z-step of 0.30 µm, were acquired using 405, 488, 552, and 638 laser lines or a white laser with optimized wavelengths at the peak of excitation for each fluorophore.

*Confocal microscopy for in vivo tissue.* Confocal microscopy was performed using an inverted Leica SP8 with ×40 or ×63 oil immersion objectives with 1.30 and 1.40 numerical aperture respectively, using LasX software. For quantification, 1024 × 1024 pixel images were acquired using a ×40 oil objective with a numerical zoom of 2, corresponding to a 0.0211 mm² final area, and 10 sections were acquired with a step of 0.30 µm. For 3D reconstruction, images were acquired using the 63x oil immersion objective, and a z-step of 0.20 µm. Deconvolution was carried out using Huygens software (v.17.10). Following deconvolution, the surface of each structure was reconstructed in 3D using Imaris software (GraphPad, Bitplane, v.9.2). Figures were made using Photoshop (Adobe, version CC).

**Live-imaging**
*Live-imaging study ex vivo.* Prior to imaging, the slices were mounted onto 35 mm glass-bottom dishes (Ibidi, BioValley) and incubated with a phenol-red free medium consisting of: 75% DMEM, 20% 1X HBSS, supplemented with $HCO_3^-$ (0.075 g/L final), HEPES Buffer (10 mM final), GlutaMax (2 mM final), penicillin–streptomycin (100 IU/mL each), 5% heat-inactivated horse serum, all from Thermo Fischer Scientific, and D-Glucose (4.5 g/L final; Sigma). Imaging was performed using a Yokogawa CSU-X1 M Spinning Disk, on an inverted Leica DMi8 microscope, with a X40 NA 1.30 oil immersion objective, using the 488 nm and 561 nm laser lines and a Hamamatsu Flash 4 LT camera. Images were acquired at 37 °C in a temperature-controlled chamber under 5% CO2 using Metamorph software (Molecular Devices), for periods of 10 min with a confocal Z-series stack of 10µm (921 × 1024 pixels, 177.8 × 197.7 µm) acquired for both GFP and mCherry every 30 s, using a Z-interval of 0.5µm between optical slices. The faint mCherry signal along Purkinje cells axons allowed to follow them during live acquisition (see Fig. S4Dii). Nodal structures were detected as the non-moving β1Nav-mCherry clusters along these axons.

*Longitudinal live-imaging study in vivo.* At 7 DPI (peak of demyelination) or 11 DPI (early phase of remyelination), the mice were anesthetized as described before and anesthesia was maintained by reinjection of Ketamin/Xylazin (respectively 11/2.5 mg/kg) when needed along the experiment. Imaging was performed in a heated chamber at 34 °C using an upright 2-photon microscope Zeiss 710 NLO with a X20 water objective (NA 1,0) and a Coherent Vision II laser. Z-series stack (1024 × 1024 pixels, 170 × 170 µm) were acquired with a Z-interval of 0.78 µm at an excitation wavelength of 940 nm every 10 or 30 min (15 to 20 images per stack). Focal lesions were identified by the visualization of microglial activation and acquisitions were made at lesion direct vicinity (<250 µm) at 7 DPI and in the remyelinating area at 11 DPI (nodal structures were selected within the lesion border). We selected nodes initially contacted by microglia for acquisitions and performed 1 h movies (one stack every 10 min) and 3-h movies (one stack every 30 min). Following imaging, the mice at 7 DPI were placed in a warming chamber until awaken and re-imaged at 11 DPI. Post-acquisition image processing was carried out using ImageJ (NIH, Bethesda, Maryland), Zen (Zeiss, 2010). Briefly, images were realigned using StackReg in ImageJ (http://bigwww.epfl.ch/thevenaz/stackreg/). Filter Median (3,0 X and Y) was used in Zen and Substract Background (100 pixels in green) in ImageJ.

**Analysis**
*Quantification of microglia-node contacts.* For the ex vivo fixed slices, using ImageJ software, the brightness and contrast of the images were adjusted and the total number of nodal structures, as well as the number of nodal structures contacted by microglial cells were quantified per field using the middle plan of each Z-series, the rest of the serie being used to confirm the nature of the contacted structure, and exclude potential granule cell axon initial segment. A contact was defined by at least one positive pixel for the microglial marker juxtaposed to at least one pixel positive for the nodal marker. Three to five images were analyzed per animal with at least four animals per condition and the mean percentage of contacted nodes per condition was calculated by doing the mean of the mean percentage of contact per animal.

For the in vivo fixed tissue study, imageJ software was used to adjust the brightness and contrast of the images and to quantify the total number of nodal structures and the number of structures contacted per field using the middle plan of each Z-series, the rest of the serie being used to confirm the nature and the contacted structure. A contact was defined by at least one positive pixel for the microglial marker juxtaposed to at least one pixel positive for the nodal marker. Three to five images were analyzed per animal with four animals per condition and the mean percentage of contact per condition was calculated by doing the mean of the mean percentage of contact per animal. For the study of IGF-1 and iNOS expression in "non contacting" and "contacting" microglial cells during remyelination, we used ImageJ software to detect IBA1 + microglial cells entirely located within the stack. Each cell was then sorted as "contacting" or "non contacting" nodes and as iNOS +/− or IGF1 +/− expressing cells.

*Quantification of microglial phenotype and remyelination.* To analyze the effect of TEA and TPA treatments on remyelination ex vivo, five images of an entire folium were acquired per condition for each animal (1900 × 1900 pixels, ~550 × 550 µm). The myelination index was calculated semi-automatically using a custom written script on ImajeJ, from Baudouin et al.[72]. Briefly, a region of interest including Purkinje cells axon (excluding soma and white matter tracks) was first selected. A mask for axonal area (Calbindin signal) and a mask for myelinated axonal area (PLP signal overlapping with Calbindin signal) were then generated, and the myelination index was calculated from the quotient of the area of the two respective masks (myelin/axon). Myelination indexes of the five images were averaged to give the mean myelination index per animal for each condition. The microglial phenotype was further analyzed by calculating the amount of Iba1+ cells expressing IGF-1 (pro-regenerative microglia) or iNOS (pro-inflammatory microglia) on the total number of Iba1+ cells (using five images per condition for each animal, 1024 × 1024 pixels, 290.9 × 290.9 µm). For each experiment, at least six animals were analyzed, with paired control and treated slices from each animal.

For the in vivo study following LPC-induced demyelination, microglial expression of IGF1 was analyzed by calculating the amount of Iba1+ IGF1+ cells on the total number of Iba1+ cells (using three images for each animal taken at the border of the remyelinating area, 1024 × 1024 pixels, 145.5 × 145.5 µm). To assess the percentage of the lesion not remyelinated, a mosaic image of the whole lesion was acquired with X20 oil objective (NA 0.75) for each animal on the section on which the lesion was the largest (enrichment in Iba1 positive cells). Tiles were stitched using LasX software and analysis performed on ImageJ software. The total area of the lesion (Activated microglia, Iba1 + signal, with high DAPI density) and the area without myelin signal (PLP) were measured. The percentage of the lesion not remyelinated was calculated by dividing the area without myelin signal by the total area of the lesion for each animal.

*Time-lapse imaging analysis.* For ex vivo time-lapse acquisitions, the field of interest was defined by selecting a node (i-e β1Nav-mCherry cluster along an axon) or an internodal area initially contacted by the tip of a microglial process. ImageJ software was used to adjust the brightness and contrast of the images of each movie and ImajeJ stabilizer Plugins was used to realign the different timeframes using the nodes as stationary reference (red channel) and subsequently applying the same

corrections to the green channel (microglia, moving). For each timepoint, the contact was defined by at least one positive pixel for the microglial marker juxtaposed to at least one pixel positive for the nodal marker. The percentage of frames with contact and the maximum number of consecutive frames with and without contact were then calculated.

To analyze the dynamics of microglial process tips initially contacting a node or not, the images were first realigned automatically and the exact coordinates of the node (or original location of tips without contacts) were measured. For each movie, the microglial process tip contacting the node and of two non-contacting microglial process tips chosen randomly were tracked manually to get their coordinates at each timepoint. For each time point, the t0 coordinates were substracted from the coordinates of the microglial process tips to get the coordinates relative to their initial position (fixed). Lastly, for each microglial tip independently, the position at t0 was set in Cartesian coordinates at (0, 0). Representation of the trajectories in Rose plots were generated using R and the distance between two successive timepoints $t_n$ and $t_{n+1}$ in the Cartesian coordinates were calculated using the following formula:

$$t_{n+1}t_n = \sqrt{(x_{tn+1} - x_{tn})^2 + (y_{tn+1} - y_{tn})^2} \quad (1)$$

For the in vivo live-imaging study of the microglia-node contact in dorsal spinal cord, image fields were chosen where we had initially a clear contact between at least one node and one microglial cell. In case of multiple initial contacts between node(s) and microglia on the first timeframe, each microglia-node pair was considered independently. ImageJ software was used to adjust the brightness and contrast of the images for each movie. The Z-series were then used to assess whether the nodal structure(s) were contacted by the microglial cell(s) or not. A contact was defined when at least one pixel positive for a microglial signal was juxtaposed to at least one pixel positive for the nodal signal. Whether the contact was maintained or lost for each microglia-node pair was assessed along each movie and the percentages of timeframes with contact per movie, as well as the longest sequence of consecutive timepoints with contact, were calculated for each microglia-node pair.

*Microglial surveillance and ramification analysis.* Movies were acquired as described above. Computation of microglia surveillance index and ramifications index were performed using custom written Mathlab script from Madry et al., available at https://github.com/AttwellLab/Microglia. Briefly, movies were registered in ImageJ and realigned using the stabilizer plugin and "substract Background" was applied with a ball size of 30 pixels. Each image within stacks was then filtered using a 3 pixels median filter. A maximum intensity orthogonal projection was made and individual cells of interest were selected by drawing a region of interest that included all process extensions along the full movie. Each cell was then manually binarized and registered as an independent file.

The index of ramification R was calculated as the ratio of the perimeter to the area normalized by the similar ratio calculated for a circle of this area and subsequently normalized with control conditions:

$$R = \left(\frac{\text{perimeter}}{\text{area}}\right) \bigg/ \left(2 \times \sqrt{\pi / \text{area}}\right) \quad (2)$$

To quantify the surveillance index, for each movie, the frame n-1 was subtracted to the frame n and two binarized movies were generated. The first one containing only the pixels explored by process extension (PE) and the second the pixels associated to process retraction (PR), the unchanged pixels being set at 0. The surveillance index S was then calculated as indicated below and subsequently normalized with control conditions:

$$S = \sum_{\text{pixels}} \text{PE} + \text{PR} \quad (3)$$

*Statistical analysis.* All statistical analysis and data visualization were performed using Prism (GraphPad, version 7) and R (R Foundation for Statistical Computing, Vienna, Austria, 2005, http://www.r-project.org, version 3.5.2).

For all the experiments, the group size and statistical tests applied are indicated in the text or in the figure legends. Detailed information is further available in the statistical analysis table (supplementary data). Graphs and data are reported as the mean ± SEM. The level of statistical significance was set at $p < 0.05$ for all tests. Asterisk denote statistical significance as follow: $*p < 0.05$, $**p < 0.01$, $***p < 0.001$, $****p < 0.0001$, ns. indicates no significance.

For the fixed ex vivo cultured slices, the treated and untreated groups were compared using two-sided paired test. Parametric tests (paired Student's $t$ tests) were used when a sample size $n \geq 6$ and the distributions passed the normality test (Shapiro–Wilk test). Otherwise, non-parametric tests were applied (Wilcoxon matched pairs tests).

For ex vivo live-imaging, when measuring the percentage of frames in contact and the maximum consecutive frames with or without contact, two-sided Mann–Whitney tests were used.

The study of the dynamics of process tips contacting a nodal structure or without contact, involved repeated measure design. The statistical analysis was thus performed with linear mixed models using lmer function from lme4 package (R software). Significance of the main effect was evaluated with the Anova function using Type II Wald $\chi^2$ tests. When necessary, to better match the model

assumptions (normality and constant variance of residuals), the data were square root transformed prior to modeling. For data representation, the estimated marginal means and model based standard errors were extracted for each condition using emmeans package (R software).

For the in vivo fixed tissue study regarding the percentage of nodes contacted, comparisons involving five conditions in a repeated measure design were conducted with linear mixed models using the lmer function in the lme4 package. Significance of the main effect was evaluated with the Anova function in the car package in R using Type II Wald $\chi^2$ tests. Two-sided Tukey's post hoc pairwise comparisons were then performed using the emmeans and pairs functions in the emmeans package.

For the in vivo live-imaging studies, as the condition of normal distribution was not fulfilled within the groups, we used non-parametric tests. The differences between the groups were estimated using the non-parametric Kruska–Wallis test followed by two-sided post hoc Dunn's test with further $p$-value adjustment by the Benjamini–Hochberg false discovery rate method.

For the in vivo fixed tissue study coupling LPC-induced demyelination with osmotic pump implantation, Grubbs' test was used to identify potential outliers ($n = 6$ in each group, distribution passed Shapiro–Wilk normality test), one outlier was removed of all the analysis in the LPC+Vehicle group ($p = 0.0316$), no outliers were detected in the group treated with LPC+TPA. The two groups were then compared using two-sided Mann–Whitney tests. For the in vivo fixed tissue study regarding the "conctacting/non contacting" iNOS/IGF1 expressing microglial cells, a compute Cochran-Mantel-Haenszel $\chi^2$ test of the null hypothesis that two nominal variables ("contact" and "protein expression") are conditionally independent in each animal was performed.

**Reporting summary**. Further information on research design is available in the Nature Research Reporting Summary linked to this article.

## Data availility
Raw source data files and detailed plasmid description are available upon request. Source data are provided with this paper.

## Code availability
The codes used are referenced in the methods.

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

## Acknowledgements

We thank Prof P.J. Brophy, University of Edinburgh, UK and Prof S. Jung, Tel Aviv, Israel for kindly providing the Thy1-Nfasc186mCherry and the CX3CR1-GFP mouse line and Dr P. Ravassard for the gift of the pTrip-Syn plasmid. We thank Prof G. Rougon and Dr F. Debarbieux, INT, France, for the in vivo spinal cord window technique, and Dr Roberta Magliozzi, University of Verona, Italy, for the immunohistostaining

technique on human post-mortem tissue. We thank Dr B. Zalc, Dr N. Sol-Foulon, and Dr B. Stankoff for insights and fruitful discussions. We thank the MS Society Tissue Bank at Imperial College London and Dr. Gveric for the provision of the MS brain samples (supported by grant 007/14 from the UK MS Society). We thank the icm.Quant imaging platform, ICM biostatistics platform (iCONICS), CELIS, electrophysiology, histology, vectorology and genotyping ICM platforms, and PhenoICMice ICM facilities. All animal work was conducted at the PHENO-ICMice facility (supported by ANR-10- IAIHU-06 and ANR-11-INBS-0011-NeurATRIS and FRM). This work was funded by INSERM, ICM, ARSEP Grants (to C.L. and A.D.), FRM fellowships (SPF20110421435, to A.D. and FDT20170437332, to M.T.), APHP and ARSEP travel grant fellowship to T.R., Prix Bouvet-Labruyère - Fondation de France (to A.D.), BBT (ICM; to A.D.), ANR JC (ANR-17-CE16-0005-01; to A.D.), and FRC (Espoir en tête, Rotary Club).

## Author contributions

A.D., R.R., T.R., and C.L. designed research. R.R., T.R., M.S.A., and A.D. performed research and analyzed data. L.R. and J.M.V. generated electron microscopy data. T.R. and A.D. designed and performed the human study. F.X.L., R.R., T.R., and A.D. designed and did the biostatistical analysis. M.T. and E.M. participated in project initiation and experimental design. A.D., R.R., T.R., and C.L. wrote the paper.

## Competing interests

The authors declare no competing interests.
