## [Peer Review File · Nature Communications]

Reviewers' Comments:

Reviewer #1:

Remarks to the Author:

In this manuscript, Ronzano et al investigate the interaction of microglia with nodes of Ranvier, demonstrating for the first time that microglia-node interactions regulate microglia activation and remyelination. This study is important as i) the mechanisms controlling remyelination are not fully understood, ii) microglia are implicated in regulating remyelination yet the mechanisms influencing their activation, and in turn remyelination, are also unclear. This is an elegant and well-performed study, which uses a wide range of techniques to assess microglia-node interaction in vivo and ex vivo (EM, immunofluorescence, live imaging, pharmacological intervention). The data is conclusive and reinforced by the convincing live imaging. This is a very novel concept and one which expands our understanding of how microglia and remyelination are regulated, and I would thus predict this study to have an important impact on the field.

With regards to the data, I have the following suggestions.

Major issues:

1. The authors show that remyelination increases the frequency of contacts between microglia and nodes. It is not clear if this is an increase in this contact, or whether it is reflecting a change in node or microglia numbers. Can the authors determine if there is a change in the number of nodes or the number of microglia, in association with the changes in contacts during remyelination? Confirming this in the TTX and TEA experiments would also be important, as a rapid change in microglia could result in the changes in the contacts.
2. Potassium channel signalling in microglia is proposed as the mechanism whereby microglia communicate at the node, as potassium channel blockers dampen the contacts. However, this could be indirect as these blockers may act on other cells too. Can the authors confirm THIK-1 expression on microglia in the explants, and determine what other cells express it, with and without remyelination? Is THIK-1 expression increased on microglia during remyelination, or are K⁺ fluxes increased?
3. The robust effects on remyelination are, if I understood the methods correctly, observed 15 hours after treatment with the inhibitors. This is incredibly fast to see an impact on remyelination – could it rather be a second wave of demyelination?
4. Microglia are quantified to interact more frequently with nodes vs internodes. Can the authors clarify how the internodes were identified in the quantifications, as the images provided have a dotted line but no bright field or myelin stain to show where the internodes are.
5. The authors show a quantification with no difference between contacts with mature vs immature nodes. It is not clear how this was determined – can the authors clarify and show representative images?
6. The iNOS staining in the brain explants is a bit spotty – this is a notoriously difficult stain to get working in explants and often requires very short fixations (10 minutes max). Can the authors either optimize this stain or provide a second readout of pro-inflammatory activation?

Minor issues:

1. The authors show that the microglia-node contacts during remyelination alter microglia activation and remyelination. Can the authors speculate as to what the function could be during development or homeostasis, when the microglia would not be inflammatory?
2. The authors show nice images of the interactions of microglia with nodes in white matter and grey matter without injury in human and mouse in Figures 1 and 2. Although a summary quantification is provided in the text, can the authors show the data in a graph as well, so the individual mice/human numbers (and potential variability) can be seen?
3. It is now understood that microglia are heterogeneous, including in remyelination. Can the authors provide a quantification of the percentage of microglia which are contacting nodes? This could potentially represent different subsets of microglia (contacting vs non contacting microglia).
4. Can the authors boost the contrast of the nodes in Fig.4 as it is difficult to see where the arrows are pointing.
5. I would recommend taking out the sentence in the discussion on IGF1+ microglia in contact with the nodes, as this is 'data not shown' and therefore either the data should be included, or this sentence removed.
6. Can the authors provide a post-hoc test for the multiple comparisons in Supplemental Figure 1,

to determine whether the 7 day perilesional contacts are decreased in stability compared to the sham?

Reviewer #2:

Remarks to the Author:

The authors demonstrate by immunohistochemical staining and immune EM techniques that processes of Iba1+ microglia can be found seemingly interacting with the axonal membrane at nodes of Ranvier. They found that nearly 30% of all nodes have such microglial contacts. The definition of what constitutes an "interaction" is critical, however, and it may be difficult to judge these proposed interactions simply by light microscopy with intracellular fluorescence. To at least distinguish random contacts from stable interactions, cells from transgenic mice expressing cell-specific fluorescent markers were observed in the spinal cord (ex vivo) over time. While that is in principle a smart idea, the chosen images and videos are not terribly convincing. From the fact that one continuously observed microglia does not visibly walk away from a single node (video 1), I would not be able to conclude underlying stable interactions. Video 2 is a little bit better, but only n=1. That the frequency of interactions decreases upon demyelination, is likely caused by the enhanced process motility of microglia, but cannot yet prove that the colocalization of microglial processes with axons marks otherwise "stable" underlying interactions. The challenging issue here is the difference between specific and the many unspecific contacts (biochemically speaking), which are likely different between myelin membranes and axonal membranes as putative microglial target structures. The attempt to invoke neuronal potassium release as a trigger of altered microglial behavior is interesting, but certainly indirect. How do the authors imagine that potassium release and the identified kinase alter the physical interactions, i.e. adhesion, of two cells? The authors should also consider the additional role of oligodendroglial paranodes that are biochemically distinct from the myelin sheath. Morphologically, I find it quite difficult to distinguish microglial interactions with nodal and paranodal structures. Overall this is an interesting first shot at the identification of microglial interactions with axons in myelinated tracts, heavily based on in vitro data and still lacking the most relevant molecular players.

Reviewer #3:

Remarks to the Author:

The research paper by Ronzano et al., reports a novel form of interaction between microglia and nodes of Ranvier. The authors performed informative anatomical studies complemented with in vivo and ex vivo imaging to show that microglia specifically interact with nodes of Ranvier and these interactions in part depend on the regulation of extracellular potassium levels. The authors also suggest that microglia participate in pro-remyelinating effects through these interactions and demyelination polarizes microglia towards a pro-inflammatory phenotype.

The anatomical analysis used to demonstrate the specific interactions between microglia and nodes of Ranvier is convincing and supports the evolving concept of compartment-specific cross-talk between neurons and microglia. I find it more difficult to interpret in vivo imaging data and to connect in vivo measurements with studies performed using organotypic slice cultures.

Nevertheless, this paper is valuable and with appropriate revisions it would make an important contribution to the field of microglia-neuron interactions or neurological diseases. The authors should consider the following specific points:

- CX3CR1GFP+/Thy1-Nfasc186mCherry double-transgenic mice may be an ideal tool for the imaging studies performed given that the specificity of the red (mCherry) signal for nodes of Ranvier is demonstrated. This should be done on perfusion fixed tissues by using immunofluorescence to colocalize Thy1-Nfasc186mCherry with AnkyrinG+ / Caspr+ staining. Since AnkyrinG is strongly expressed at sites of axon initial segments too, it must be demonstrated that the authors assessed interactions between nodes of Ranvier and microglia during the different imaging studies they performed. Multi-color immunofluorescence to validate the specificity of reporter protein expression by nodes of Ranvier for cerebellar slices transduced to express β 1NaVmCherry is also required.

- I find in vivo imaging conditions suboptimal to make firm conclusions about the interactions between microglia and nodes of Ranvier. In the methods, the authors describe that they took one stack no more frequently than every 10 minutes: „We selected nodes initially contacted by microglia for acquisitions and performed 1 hour movies (one stack every 10 minutes) and 3-hour movies (onestack every 30 minutes)“. This imaging protocol does not allow proper assessment of microglia process dynamics and thus at least a set of validation in vivo imaging studies should be provided to make these results sound by capturing process responses at least at 1-2 image/min frequency in different Z planes.

- „We detected that both microglial processes and soma are motile over long periods of time.“ Please clarify this statement by adding details on the speed/dislocation of both cell bodies and processes of microglia. In the healthy brain and spinal cord tissue, microglial cell bodies are relatively steady compared to highly motile processes. Do the authors suggest that microglial cell bodies were similarly motile to processes in these imaging studies? Suboptimal time-resolution of in vivo imaging may also explain why microglial process dynamics has not been appropriately visualized.

- „...we observed that microglia-node contacts were maintained along the vast majority of 1 hour movies, as well as 3 hour movies“. How was this measured, were data originate from a given time point or averaged from time-lapse measurements? What was the average lifetime of microglia – nodes of Ranvier contacts? Were the same nodes recontacted by microglial processes or cell bodies over a longer period of time? It would be generally useful to depict the lifetime of contacts and their changes in demyelination, since it is difficult to imagine that such a small structure is contacted steadily for prolonged time periods by otherwise motile microglial structures. Low sampling rate may also explain the (almost two-fold) difference in contact frequency between sham mice at 7 DPI and 11DPI.

- Concerning the experiments performed on organotypic slices, the authors should consider that the interpretation the results of pharmacological interventions used may be largely influenced by the fact that basic microglial phenotypes and responses are different compared to the in vivo situation. For example, P2Y12R is known to be markedly downregulated ex vivo. Therefore, the authors should investigate P2Y12 levels using immunostaining in organotypic slices to strengthen their conclusions. In addition, the effect of PSB0739 may be short-lived. In vivo verification of such conclusions would therefore be also important. For example, did the authors investigate whether the course of demyelination and/or the association between microglia and nodes of Ranvier show alterations in Cx3CR1 KO or P2Y12 KO mice?

- How did TTX and Apamin altered microglial process dynamics?

- Slice culture studies using TEA and TPA on remyelination are interesting. However, a wide-spectrum potassium inhibitor, like TEA is expected to alter the activity of glial cells (astrocytes and microglia) in addition to neurons. Any statements made here could only be interpreted if the effect of TEA on microglial activity (e.g. membrane potential) and process motility is assessed. It is not sufficient to compare nodes and internodes or to track process velocity only in TEA treated slices. Did TEA treatment impact on microglial process dynamics compared to control slices?

- Neither of the interventions used ex vivo are microglia-specific. General interference with potassium homeostasis is expected to impact on several complex processes, including remyelination. Do the authors have in vivo data showing that microglia manipulation alters the course of remyelination? This would be far more convincing.

POINT-BY-POINT RESPONSE TO THE REVIEWERS' COMMENTS

We first would like to thank the reviewers for their useful comments and suggestions, which allowed us to improve our work. A point-by-point response to their questions and concerns can be found below. The modifications are highlighted in the text of the manuscript.

Reviewer #1:

In this manuscript, Ronzano et al investigate the interaction of microglia with nodes of Ranvier, demonstrating for the first time that microglia-node interactions regulate microglia activation and remyelination. This study is important as i) the mechanisms controlling remyelination are not fully understood, ii) microglia are implicated in regulating remyelination yet the mechanisms influencing their activation, and in turn remyelination, are also unclear. This is an elegant and well-performed study, which uses a wide range of techniques to assess microglia-node interaction *in vivo* and *ex vivo* (EM, immunofluorescence, live imaging, pharmacological intervention). The data is conclusive and reinforced by the convincing live imaging. This is a very novel concept and one which expands our understanding of how microglia and remyelination are regulated, and I would thus predict this study to have an important impact on the field.

Major issues:

1. The authors show that remyelination increases the frequency of contacts between microglia and nodes. It is not clear if this is an increase in this contact, or whether it is reflecting a change in node or microglia numbers. Can the authors determine if there is a change in the number of nodes or the number of microglia, in association with the changes in contacts during remyelination? Confirming this in the TTX and TEA experiments would also be important, as a rapid change in microglia could result in the changes in the contacts.

*This is a very important point. The increased frequency of microglia-node contacts during remyelination *in vivo* (Figure 3E) is not associated with significant changes in microglia or node numbers (Figure Rev, not included in the paper for sake of space). Furthermore, normalization by microglial and node numbers (statistical analysis table, line 3) does no impact the results regarding the frequencies of contact.*

Furthermore, neither TTX nor TEA treatment lead to a significant variation of the number of microglial cells (Figure Rev-B and C respectively) or nodes (E and F respectively).

2. Potassium channel signaling in microglia is proposed as the mechanism whereby microglia communicate at the node, as potassium channel blockers dampen the contacts. However, this could be indirect as these blockers may act on other cells too. Can the authors confirm THIK-1 expression on microglia in the explants, and determine what other cells express it, with and without remyelination? Is THIK-1 expression increased on microglia during remyelination, or are K⁺ fluxes increased?

*To address these important questions, as there is no available THIK-1 antibody working to detect this protein, we studied the expression of *kcnq13* (gene encoding THIK-1) using an RNAscope approach on myelinated and remyelinating CX3CR1^{GFP/+} organotypic slices (Figure S9). We show that *kcnq13* mRNA is expressed in microglia in both myelinated and remyelinating tissues, while it is not expressed in Purkinje cells.*

This confirms RNAseq results previously obtained in mouse cerebellum, showing *kcnq13* expression in microglia, but not in neurons, astrocytes and oligodendroglial lineage cells (Carter et al., 2018). RNAscope is not a quantitative method, we thus could not address whether there is an increased expression in remyelinating slices. It is however described that in human tissue, *kcnq13* mRNA is expressed in both control and MS samples, with a similar rate of expression, and possibly a tendency to an increased expression in MS (Jäkel et al., 2019).

Regarding K⁺ flux in remyelination, it is described that a transient neuronal hyperactivity can be observed in the first phasis of remyelination (Bacmeister et al., 2020), suggesting K⁺ efflux might be increased at that time. This could strengthen microglia-node interaction in remyelination, though we cannot exclude that other signals may also be at play. This will be the scope of further studies.

3. The robust effects on remyelination are, if I understood the methods correctly, observed 15 hours after treatment with the inhibitors. This is incredibly fast to see an impact on remyelination – could it rather be a second wave of demyelination?

In the organotypic culture model, we treat the slices with the demyelinating agent LPC, overnight between DIV6 and 7. The medium is then changed for a culture medium without LPC. The peak of demyelination (DIV8-9) leads to a total demyelination of Purkinje cells axons (apart in the central white matter tracts, an area which we exclude from our analysis) and the onset of remyelination occurs at DIV10 (when we performed the remyelination analysis). As the timing of demyelination and remyelination is highly reproducible, we are confident that what we observe cannot be a second wave of demyelination. A new demyelination wave would further lead to myelin debris accumulation, which we do not observe.

4. Microglia are quantified to interact more frequently with nodes vs internodes. Can the authors clarify how the internodes were identified in the quantifications, as the images provided have a dotted line but no bright field or myelin stain to show where the internodes are.

We apologize, as this was not clearly explained in the manuscript. As shown in Figure S4Dii, amplifying the faint staining of β 1Nav-mCherry along the axon (red channel) in live acquisition was sufficient to visualize the axon. This is now clarified in the methods.

5. The authors show a quantification with no difference between contacts with mature vs immature nodes. It is not clear how this was determined – can the authors clarify and show representative images?

We apologize, as this was not stated clearly enough in the manuscript. We have now added schematics and images to clarify how mature and immature nodal structures can be distinguished (Figure S3A).

6. The iNOS staining in the brain explants is a bit spotty – this is a notoriously difficult stain to get working in explants and often requires very short fixations (10 minutes max). Can the authors either optimize this stain or provide a second readout of pro-inflammatory activation?

We agree with Reviewer 1 that iNOS is a difficult stain to get working in explants. We thus performed additional stainings with IGF1 antibody, IGF1 being an established marker for pro-regenerative microglia, and quantified the number of IGF1 positive microglial cells in remyelinating slices. As shown in Figures 7 (*ex vivo*) and 8 (*in vivo*), we observed a significant decrease of IGF1 positive microglia following TEA or TPA treatments (7D-E and 7H-I respectively *ex vivo*, and 8B-C *in vivo*), suggesting an alteration of the microglial switch, which is in line with the results previously observed with iNOS. We also improved iNOS staining, (now Figure S9E-H).

Minor issues:

1. The authors show that the microglia-node contacts during remyelination alter microglia activation and remyelination. Can the authors speculate as to what the function could be during development or homeostasis, when the microglia would not be inflammatory?

Recently published articles (Djannatian et al., 2021; Hughes and Appel, 2020) suggest that microglia can participate in myelin phagocytosis during development and could thus participate in myelin pattern regulation along axons. In line with these data, we can hypothesize that microglia-node interaction could participate in the regulation of this process. Furthermore, as mentioned in the discussion, in “*The node of Ranvier: a neuron-glia communication hub?*” subpart, it is known that perinodal astrocytes can modulate nodal length by secreting Serpine2, which then modulates paranodal loops attachment to the axonal membrane (Dutta et al., 2018). Microglial cells can express this factor, thus microglial read-out of neuronal activity at nodes could also participate in this process. More generally, perceiving neuronal physiological status at nodes during development and homeostasis could participate in microglial role in supporting neuronal survival and physiology. Although this is an exciting field, we cannot include this discussion in the paper due to space constraints.

2. The authors show nice images of the interactions of microglia with nodes in white matter and grey matter without injury in human and mouse in Figures 1 and 2. Although a summary quantification is provided in the text, can the authors show the data in a graph as well, so the individual mice/human numbers (and potential variability) can be seen?

These data are now presented in Figure S1.

3. It is now understood that microglia are heterogeneous, including in remyelination. Can the authors provide a quantification of the percentage of microglia which are contacting nodes? This could potentially represent different subsets of microglia (contacting vs non contacting microglia).

In our different experiments, all microglia were contacting nodes in control myelinated mouse tissues. We however observed a small proportion of microglial cells that did not contact nodes in remyelinating condition in mouse spinal cords (Figure S9). Regarding human tissues, the detailed study of microglia-node interaction in normal and MS tissues is presently ongoing and will be the scope of a further publication.

4. Can the authors boost the contrast of the nodes in Fig.4 as it is difficult to see where the arrows are pointing.

We have modified the contrast and brightness of the images in Figure 4 to make the nodes more visible.

5. I would recommend taking out the sentence in the discussion on IGF1+ microglia in contact with the nodes, as this is 'data not shown' and therefore either the data should be included, or this sentence removed.

These *in vivo* data are now presented in Figure S9I-K.

6. Can the authors provide a post-hoc test for the multiple comparisons in Supplemental Figure 1, to determine whether the 7 day perilesional contacts are decreased in stability compared to the sham?

All post-hoc tests are presented in the statistical analysis table. Regarding contact stability by live-imaging, the reduction of stability is clearly significant for 1-hour movies between perilesional and sham DPI7 ($p < 0,0001$), but is only a tendency for 3-hour movies ($p = 0,0805$). We rephrased the reference to the three-hour movies in the text to clarify this point.

Reviewer #2:

The authors demonstrate by immunohistochemical staining and immune EM techniques that processes of Iba1+ microglia can be found seemingly interacting with the axonal membrane at nodes of Ranvier. They found that nearly 30% of all nodes have such microglial contacts. The definition of what constitutes an "interaction" is critical, however, and it may be difficult to judge these proposed interactions simply by light microscopy with intracellular fluorescence.

As mentioned by the Reviewer, we first confirmed that the contacts detected by immunohistostainings in Figure 1 were direct contacts by performing an electron microscopy study. Having shown that they corresponded indeed to direct contact between the axolemma and microglia, we then defined a microglial-node contact such as there was at least a "microglia" pixel directly touching a "nodal" pixel. This definition was used for our quantification study by confocal microscopy (with an approximative resolution of 200nm), light microscopy being classically used to study microglial interaction with neuron sub-compartment following electronic microscopy validation (Cserép et al., 2020).

To at least distinguish random contacts from stable interactions, cells from transgenic mice expressing cell-specific fluorescent markers were observed in the spinal cord (*ex vivo*) over time. While that is in principle a smart idea, the chosen images and videos are not terribly convincing.

As described in the methods and mentioned in Figures 3 and S2 legends, the spinal cord live-imaging study (one-hour and three-hour movies) was actually performed *in vivo* on anesthetized mice with a spinal cord window, using 2-photon microscopy (technique derived from (Fenrich et al., 2012)). To further clarify this point we added "*in vivo*" acquisitions in the legends of Movies 1 to 4. Thus, we agree that the movies may not always be optimal, but this type of live-imaging is extremely demanding, due to the small size of the axonal domains studied, the presence of dense myelin, which a limiting factor for deep-imaging *in vivo*, and the fact that animal is breathing, which can affect the quality of the imaging.

From the fact that one continuously observed microglia does not visibly walk away from a single node (video 1), I would not be able to conclude underlying stable interactions. Video 2 is a little bit better, but only $n=1$.

We present only examples of videos representative of the different contexts studied, but, for 1-hour movies (Figure 3K-L), there were n=25 to 32 microglia-nodes pairs imaged *in vivo* depending on the condition considered and regarding the three-hour movies, which are more challenging to take (Figure S2D-E), n=9 to 16 pairs imaged *in vivo* depending on the condition considered. These numbers of microglia-node pairs analyzed for the quantifications are shown on the bars of the graphs, as described in the Figures Legends.

That the frequency of interactions decreases upon demyelination, is likely caused by the enhanced process motility of microglia, but cannot yet prove that the colocalization of microglial processes with axons marks otherwise "stable" underlying interactions.

The frequency of interaction (fixed tissue) is not significantly affected in perilesional tissue at the peak of demyelination (Figure 3E). The occurrence of the contact along time between a given microglia-node pair is however affected in the spinal cord *in vivo* in the perilesional tissue at the peak of demyelination (as shown by the quantifications, Figure 3K-L, and as illustrated on Figure S2B and movie 3) compared to control condition, with less timeframes in the movie where we could observe the microglial contact at the node. We agree that "stable" is not the adequate term, which is now replaced in the text, in this context, by the notion of durable or long-lasting interaction. We hope this answers the concern of the reviewer.

The challenging issue here is the difference between specific and the many unspecific contacts (biochemically speaking), which are likely different between myelin membranes and axonal membranes as putative microglial target structures. The attempt to invoke neuronal potassium release as a trigger of altered microglial behavior is interesting, but certainly indirect. How do the author s imagine that potassium release and the identified kinase alter the physical interactions, i.e. adhesion, of two cells?

There must be a misunderstanding, as we are not identifying any kinase as implicated in neuronal potassium signaling here. Indeed, using TEA and TPA, we targeted either axonal potassium channels or a 2-Pore potassium channel expressed by microglia, THIK-1 (Madry et al., 2018), respectively. In both conditions, this K⁺ channel blockade results in reduced microglia-node interaction. Our hypothesis is that potassium release at the node (linked to neuronal activity; blocked by TEA) may lead to reduced microglial process dynamics at the node (mediated by microglial channel THIK 1) and thus to the fact that microglia preferentially interact with the nodes compared to elsewhere along the axons.

The authors should also consider the additional role of oligodendroglial paranodes that are biochemically distinct from the myelin sheath. Morphologically, I find it quite difficult to distinguish microglial interactions with nodal and paranodal structures.

We thank the reviewer for this interesting question. The fact that the microglial contact that we observed at nodes extends to the first paranodal loops is very exciting indeed. One could hypothesize that it is driven by a potential leak of the potassium released by the juxtaparanodal potassic channels Kv1.1 and Kv1.2, and this is a question we hope to address in the future.

Overall this is an interesting first shot at the identification of microglial interactions with axons in myelinated tracts, heavily based on *in vitro* data and still lacking the most relevant molecular players.

We thank the reviewer for his interest for the question assessed by our work. We however have to respectfully disagree with him, as our work includes both *in vivo* and *ex vivo* data. Although we agree that we still lack complete deciphering of these microglia-node interactions, our study has further been strengthened by new sets of *in vivo* data (Figure S5C-D, Figure 8 and figure S9I-K).

Reviewer #3:

The research paper by Ronzano et al., reports a novel form of interaction between microglia and nodes of Ranvier. The authors performed informative anatomical studies complemented with *in vivo* and *ex vivo* imaging to show that microglia specifically interact with nodes of Ranvier and these interactions in part depend on the regulation of extracellular potassium levels. The authors also suggest that microglia participate in pro-remyelinating effects through these interactions and demyelination polarizes microglia towards a pro-inflammatory phenotype.

The anatomical analysis used to demonstrate the specific interactions between microglia and nodes of Ranvier is convincing and supports the evolving concept of compartment-specific cross-talk between neurons and microglia. I find it more difficult to interpret *in vivo* imaging data and to connect *in vivo* measurements with studies performed using organotypic slice cultures. Nevertheless, this paper is valuable and with appropriate revisions it would make an important contribution to the field of microglia-neuron interactions or neurological diseases.

The authors should consider the following specific points:

- CX3CR1GFP+/Thy1-Nfasc186mCherry double-transgenic mice may be an ideal tool for the imaging studies performed given that the specificity of the red (mCherry) signal for nodes of Ranvier is demonstrated. This should be done on perfusion fixed tissues by using immunofluorescence to colocalize Thy1-Nfasc186mCherry with AnkyrinG⁺ / Caspr⁺ staining. Since AnkyrinG is strongly expressed at sites of axon initial segments too, it must be demonstrated that the authors assessed interactions between nodes of Ranvier and microglia during the different imaging studies they performed. Multi-color immunofluorescence to validate the specificity of reporter protein expression by nodes of Ranvier for cerebellar slices transduced to express β 1NaVmCherry is also required.

The data requested are now presented in new Figure 3H, which shows that Nfasc186mCherry is restricted to nodes in mouse dorsal spinal cord tissue (we focus our live-imaging study on dorsal white matter tracts, which are deprived of AIS), and new Figure S4A-B for the *ex vivo* expression of β 1NaVmCherry.

- I find *in vivo* imaging conditions suboptimal to make firm conclusions about the interactions between microglia and nodes of Ranvier. In the methods, the authors describe that they took one stack no more frequently than every 10 minutes: „We selected nodes initially contacted by microglia for acquisitions and performed 1 hour movies (one stack every 10 minutes) and 3-hour movies (onestack every 30 minutes)”. This imaging protocol does not allow proper assessment of microglia process dynamics and thus at least a set of validation *in vivo* imaging studies should be provided to make these results sound by capturing process responses at least at 1-2 image/min frequency in different Z planes.

We agree with the reviewer that to properly study the behavior of microglial most dynamic processes and assess the stability of their contact at nodes, shorter movies with 1 image per minute (with the acquisition of a Z-stack) are needed.

Thus, we first replaced any mention of “stable” interaction between microglia and nodes referring to 1-h and 3-hour movies with the terms “long-lasting” or “durable”, to underline that the interaction perdured without necessarily being maintained permanently. Secondly, we performed a set of new experiments *in vivo* in mouse dorsal spinal cord by 2-Photon microscopy. We adapted the acquisition parameters to make faster acquisitions of microglia-nodes pairs in myelinated tissue *in vivo* (10 minute movies, one stack acquired per minute, 18 pairs from 9 movies, n=4 mice, one representative example: Movie Rev. Scale bar: 10 μ m). In all the cases, we observed 100% frames with contact, however, only contacts established with cell soma or main microglial process could be observed, as the tips of thin microglial processes could not be visualized with these acquisition parameters. This is why the study of the dynamics at nodes of the tips of thin microglial processes was performed *ex vivo* in the first place, as it allows for higher spatial and time resolution. This has now been clarified in the text of the manuscript.

- „We detected that both microglial processes and soma are motile over long periods of time.” Please clarify this statement by adding details on the speed/dislocation of both cell bodies and processes of microglia. In the healthy brain and spinal cord tissue, microglial cell bodies are relatively steady compared to highly motile processes. Do the authors suggest that microglial cell bodies were similarly motile to processes in these imaging studies? Suboptimal time-resolution of *in vivo* imaging may also explain why microglial process dynamics has not been appropriately visualized.

We apologize as this sentence was not clear indeed. We changed it to “We confirmed that microglia are dynamic cells, with motile processes, and that their whole morphology can vary over long periods of time (Movie1)” hoping this makes clearer that soma and processes do not have similar dynamics.

- „...we observed that microglia-node contacts were maintained along the vast majority of 1 hour movies, as well as 3 hour movies”. How was this measured, were data originate from a given time point or averaged from time-lapse measurements? What was the average lifetime of microglia – nodes of Ranvier contacts? Were the same nodes recontacted by microglial processes or cell bodies over a longer period of time? It would be generally useful to depict the lifetime of contacts and their changes in demyelination, since it is difficult to imagine that such a small structure is contacted steadily for prolonged time periods by otherwise motile microglial structures. Low sampling rate may also explain the (almost two-fold) difference in contact frequency between sham mice at 7 DPI and 11DPI.

For 1-hour and 3-hour *in vivo* live imaging movies, we searched for a microglial cell initially contacting a nodal structure (which we define as microglia-node “pair”) and then imaged these “pairs” every 10 minutes and 30 minutes respectively, to assess whether a given microglia could recurrently be found contacting a given node along time and what was the occurrence of this contact (number of timeframes with contact) and the maximum number of consecutive timeframes where we would observe this contact during a movie. These data, which we quantified for each microglia-node pair, are presented in figures 3K and L (one-hour movie) and S2D and E (three-hour movie), each dot corresponding to a microglial-node pair. They show that a given node is indeed contacted on most of the timeframes by its associated microglia, apart in the perilesional area at the peak of demyelination, where the number of timeframes with contact decreases (as well as the maximum consecutive timeframes with contact). Regarding the very motile, thin microglial processes, we performed an equivalent study using our *ex vivo* model to have an adequate spatial and time resolution. The data are presented in Figure 4.

- Concerning the experiments performed on organotypic slices, the authors should consider that the interpretation the results of pharmacological interventions used may be largely influenced by the fact that basic microglial phenotypes and responses are different compared to the *in vivo* situation. For example, P2Y12R is known to be markedly downregulated *ex vivo*. Therefore, the authors should investigate P2Y12 levels using immunostaining in organotypic slices to strengthen their conclusions. In addition, the effect of PSB0739 may be short-lived. *In vivo* verification of such conclusions would therefore be also important. For example, did the authors investigate whether the course of demyelination and/or the association between microglia and nodes of Ranvier show alterations in Cx3CR1 KO or P2Y12 KO mice?

As suggested, we have now included a P2Y12R immunohistostaining of cultured cerebellar slices showing this receptor is indeed expressed in microglial cells in our model (Figure S6A). Regarding the P2Y12 KO mice, we did not have this mouse line in house and could not import it due to the present Covid restrictions. However, as there may have been compensation by other P2Y receptors in this model, the results may have been inconclusive. On the other hand, we were able to perform the requested experiments *in vivo* with CX3CR1 KO mice. The results, confirming that this pathway is not required for microglia-node of Ranvier interaction, are presented Figure S5C-D.

- How did TTX and Apamin alter microglial process dynamics?

We thank the reviewer for raising this point. The data showing that TTX and Apamin do not alter microglial process dynamics are now presented in figure S8.

- Slice culture studies using TEA and TPA on remyelination are interesting. However, a wide-spectrum potassium inhibitor, like TEA is expected to alter the activity of glial cells (astrocytes and microglia) in addition to neurons. Any statements made here could only be interpreted if the effect of TEA on microglial activity (e.g. membrane potential) and process motility is assessed. It is not sufficient to compare nodes and internodes or to track process velocity only in TEA treated slices. Did TEA treatment impact on microglial process dynamics compared to control slices?

We now present data regarding this question in Figure S8, showing that microglial morphology and microglial process dynamics are not altered following TEA treatment.

- Neither of the interventions used *ex vivo* are microglia-specific. General interference with potassium homeostasis is expected to impact on several complex processes, including remyelination. Do the authors have *in vivo* data showing that microglia manipulation alters the course of remyelination? This would be far more convincing.

It has already been shown *in vivo* in mouse that altering the microglial switch from pro-inflammatory to pro-regenerative microglia impairs remyelination (El Behi et al., 2017; Miron et al., 2013).

To strengthen our *ex vivo* data regarding the impact of microglia-node interaction in microglial switch process and remyelination with *in vivo* data, we first tried to import THIK-1 KO mouse line, but this was unfortunately delayed due to the present Covid situation.

We thus had to choose another approach, and used mini-osmotic pump to deliver TPA at the onset of remyelination above a focal lesion induced by LPC injection in mouse dorsal spinal cord. We have added in the manuscript these *in vivo* data, which corroborate our previous results, by showing that, after a focal demyelination in mouse cord, perturbing microglia-node interaction locally using TPA leads to a reduced number of IGF1+ pro-regenerative cells and a decreased remyelination (Figure 8).

References

- Bacmeister CM, Barr HJ, McClain CR, Thornton MA, Nettles D, Welle CG, Hughes EG. 2020. Motor learning promotes remyelination via new and surviving oligodendrocytes. *Nat Neurosci*. doi:10.1038/s41593-020-0637-3
- Carter RA, Bihannic L, Rosencrance C, Hadley JL, Tong Y, Phoenix TN, Natarajan S, Easton J, Northcott PA, Gawad C. 2018. A Single-Cell Transcriptional Atlas of the Developing Murine Cerebellum. *Curr Biol* **28**:2910-2920.e2. doi:10.1016/j.cub.2018.07.062
- Cserép C, Pósfai B, Lénárt N, Fekete R, László ZI, Lele Z, Orsolits B, Molnár G, Heindl S, Schwarcz AD, Ujvári K, Környei Z, Tóth K, Szabadits E, Sperlágh B, Baranyi M, Csiba L, Hortobágyi T, Maglóczky Z, Martinecz B, Szabó G, Erdélyi F, Szipőcs R, Tamkun MM, Gesierich B, Duering M, Katona I, Liesz A, Tamás G, Dénes Á. 2020. Microglia monitor and protect neuronal function through specialized somatic purinergic junctions. *Science (80-)* **367**:528–537. doi:10.1126/science.aax6752
- Djannatian M, Weikert U, Safaiyan S, Wrede C, Kislinger G, Ruhwedel T, Campbell DS, van Ham T, Schmid B, Hegermann J, Möbius W, Schifferer M, Simons M. 2021. Myelin biogenesis is associated with pathological ultrastructure that is resolved by microglia during development 2 3 4. *bioRxiv* 2021.02.02.429485.
- Dutta DJ, Woo DH, Lee PR, Pajevic S, Bukalo O, Huffman WC, Wake H, Basser PJ, SheikhBahaei S, Lazarevic V, Smith JC, Fields RD. 2018. Regulation of myelin structure and conduction velocity by perinodal astrocytes. *Proc Natl Acad Sci U S A* **115**:11832–11837. doi:10.1073/pnas.1811013115
- El Behi M, Sanson C, Bachelin C, Guillot-Noël L, Fransson J, Stankoff B, Maillart E, Sarrazin N, Guillemot V, Abdi H, Cournu-Rebeix I, Fontaine B, Zujovic V. 2017. Adaptive human immunity drives remyelination in a mouse model of demyelination. *Brain* **140**:967–980. doi:10.1093/brain/awx008
- Fenrich KK, Weber P, Hocine M, Zalc M, Rougon G, Debarbieux F. 2012. Long-term in vivo imaging of normal and pathological mouse spinal cord with subcellular resolution using implanted glass windows. *J Physiol* **590**:3665–3675. doi:10.1113/jphysiol.2012.230532
- Hughes AN, Appel B. 2020. Microglia phagocytose myelin sheaths to modify developmental myelination. *Nat Neurosci* **215**:41–47. doi:10.1038/s41593-020-0654-2
- Jäkel S, Agirre E, Mendanha Falcão A, van Bruggen D, Lee KW, Knuesel I, Malhotra D, Ffrench-Constant C, Williams A, Castelo-Branco G. 2019. Altered human oligodendrocyte heterogeneity in multiple sclerosis. *Nature* **566**:543–547. doi:10.1038/s41586-019-0903-2
- Madry C, Kyrargyri V, Arancibia-Cárcamo IL, Jolivet R, Kohsaka S, Bryan RM, Attwell D. 2018. Microglial Ramification, Surveillance, and Interleukin-1 β Release Are Regulated by the Two-Pore Domain K⁺ Channel THIK-1. *Neuron* **97**:299-312.e6. doi:10.1016/j.neuron.2017.12.002
- Miron VE, Boyd A, Zhao JW, Yuen TJ, Ruckh JM, Shadrach JL, Van Wijngaarden P, Wagers AJ, Williams A, Franklin RJM, Ffrench-Constant C. 2013. M2 microglia and macrophages drive oligodendrocyte differentiation during CNS remyelination. *Nat Neurosci* **16**:1211–1218. doi:10.1038/nn.3469

Reviewers' Comments:

Reviewer #1:

Remarks to the Author:

The authors have put in considerable effort to address my issues from the first submission and the paper has been very much improved. With regards to the new data, I have 2 minor suggestions:

1) The authors have now provided some images of microglia expressing the THIK1 gene *kcnq13* in myelinating and remyelinating conditions. The images are beautiful, but just of one cell in each condition, so it would be useful to have alongside the quantification of the number or percentage of microglia that are positive for *kcnq13* - to better understand which cells can respond to the K⁺ fluxes which are proposed to underlie the microglial response to activity during remyelination.

2) Line 137 needs some editing, the sentence with 'which was not due to microglial cell or numbers' should read 'which was not due to changes in microglial or node numbers'.

Reviewer #2:

Remarks to the Author:

The authors have addressed all our questions and I should be happy with the manuscript as is. What I am still missing is convincing evidence that the type of interaction the authors point out is specific and functional. Microglial cells happen to have motile processes that explore the environment and therefore are likely to interact (also) with axonal membranes at the node of Ranvier, in fact it is unavoidable. But may be that is too much of an academic debate right now and awaits experiments in which these contacts are experimentally blocked. I have no further objections if the authors tone down these claims in their discussion and abstract to include the points above.

Reviewer #3:

Remarks to the Author:

The authors have responded appropriately to my comments. This is a great paper.

POINT-BY-POINT RESPONSE TO THE REVIEWERS' COMMENTS

We first would like to thank again the reviewers for their useful comments and suggestions, which allowed us to improve our work. A point-by-point response to their questions and comments can be found below. The modifications are highlighted in the text of the manuscript.

Reviewer #1 (Remarks to the Author):

The authors have put in considerable effort to address my issues from the first submission and the paper has been very much improved.

We thank the reviewer for this comment and are glad that he/she appreciated the additional sets of data.

With regards to the new data, I have 2 minor suggestions:

1) The authors have now provided some images of microglia expressing the THK1 gene *kcnq13* in myelinating and remyelinating conditions. The images are beautiful, but just of one cell in each condition, so it would be useful to have alongside the quantification of the number or percentage of microglia that are positive for *kcnq13* - to better understand which cells can respond to the K⁺ fluxes which are proposed to underlie the microglial response to activity during remyelination.

*The percentage of microglia that are positive for *kcnk13* has now been quantified in the normal and remyelinating contexts and these quantifications have been added to the text, in the corresponding section.*

2) Line 137 needs some editing, the sentence with 'which was not due to microglial cell or numbers' should read 'which was not due to changes in microglial or node numbers'.

This has been changed in the text.

Reviewer #2 (Remarks to the Author):

The authors have addressed all our questions and I should be happy with the manuscript as is.

We thank the reviewer for his/her comment and are glad that he/her is satisfied by our answer and additional sets of data.

What I am still missing is convincing evidence that the type of interaction the authors point out is specific and functional. Microglial cells happen to have motile processes that explore the environment and therefore are likely to interact (also) with axonal membranes at the node of Ranvier, in fact it is unavoidable. But maybe that is too much of an academic debate right now and awaits experiments in which these contacts are experimentally blocked. I have no further objections if the authors tone down these claims in their discussion and abstract to include the points above.

We indeed observed that motile microglial processes contact the nodes, and showed preferential interaction of these processes at nodes compared to elsewhere along the axons, an effect which was abolished when we inhibited the interaction using TEA (Figure 4 and Figure 6D-I).

These data, together with the changes of microglial phenotype and remyelination (Figure 7 and 8), are important points to suggest specificity and role of this microglia-node interaction.

However, we take into consideration the referee's concern, and : i) clarify the text on these points; ii) added a sentence in the abstract and discussion to highlight the need for a further deciphering of mechanisms and role of these interactions.

Reviewer #3 (Remarks to the Author):

The authors have responded appropriately to my comments. This is a great paper.

We thank the reviewer for this comment and are glad that he/she appreciated our work.